# Rab29 activation of the Parkinson's disease-associated LRRK2 kinase

Elena Purlyte[1,†], Herschel S Dhekne[2,†], Adil R Sarhan[1] (iD), Rachel Gomez[2], Pawel Lis[1], Melanie Wightman[1], Terina N Martinez[3], Francesca Tonelli[1,†,*] (iD), Suzanne R Pfeffer[2,**] (iD) & Dario R Alessi[1,***] (iD)

## Abstract

Parkinson's disease predisposing LRRK2 kinase phosphorylates a group of Rab GTPase proteins including Rab29, within the effector-binding switch II motif. Previous work indicated that Rab29, located within the PARK16 locus mutated in Parkinson's patients, operates in a common pathway with LRRK2. Here, we show that Rab29 recruits LRRK2 to the *trans*-Golgi network and greatly stimulates its kinase activity. Pathogenic LRRK2 R1441G/C and Y1699C mutants that promote GTP binding are more readily recruited to the Golgi and activated by Rab29 than wild-type LRRK2. We identify conserved residues within the LRRK2 ankyrin domain that are required for Rab29-mediated Golgi recruitment and kinase activation. Consistent with these findings, knockout of Rab29 in A549 cells reduces endogenous LRRK2-mediated phosphorylation of Rab10. We show that mutations that prevent LRRK2 from interacting with either Rab29 or GTP strikingly inhibit phosphorylation of a cluster of highly studied biomarker phosphorylation sites (Ser910, Ser935, Ser955 and Ser973). Our data reveal that Rab29 is a master regulator of LRRK2, controlling its activation, localization, and potentially biomarker phosphorylation.

**Keywords** Golgi; GTPase; PARK genes; phosphorylation; Rab10
**Subject Categories** Membrane & Intracellular Transport; Molecular Biology of Disease; Post-translational Modifications, Proteolysis & Proteomics
The EMBO Journal (2018) 37: 1–18

## Introduction

Autosomal dominant missense mutations within the leucine-rich repeat protein kinase 2 (LRRK2) gene predispose to Parkinson's disease (Paisan-Ruiz *et al*, 2004; Zimprich *et al*, 2004). Mutations in

LRRK2 account for ~5% of familial Parkinson's, and are observed in ~1% of sporadic Parkinson's patients, making LRRK2 one of the most commonly mutated genes linked to Parkinson's disease (Simon-Sanchez *et al*, 2009). LRRK2 is a large, multi-domain protein kinase consisting of an armadillo repeat domain (residues 150–510), an ankyrin domain (residues 690–860), leucine-rich repeats (residues 984–1278), a ROC-type GTPase domain (residues 1335–1510) that closely resembles a Rab GTPase and is associated with a COR domain (C-terminal of Roc, residues 1511–1878), a serine/threonine protein kinase domain (residues 1879–2138), and a WD40 repeat-containing domain (residues 2142–2496). The most common pathogenic mutation lies within the catalytic domain (G2019S) and increases kinase activity, suggesting that LRRK2 inhibitors might offer therapeutic benefit for Parkinson's disease (Greggio *et al*, 2006; Ozelius *et al*, 2006; Smith *et al*, 2006; Jaleel *et al*, 2007; Hatcher *et al*, 2017).

Members of the Rab GTPase family, including Rab8A, Rab10, and Rab29 (also known as RAB7L1), are substrates for LRRK2 (Steger *et al*, 2016). Recent work has defined a subset of 14 Rab proteins (Rab3A/B/C/D, Rab5A/B/C, Rab8A/B, Rab10, Rab12, Rab29, Rab35, and Rab43) that are potential direct substrates for LRRK2 (Steger *et al*, 2017). The LRRK2 phosphorylation site (Thr72 for Rab8A and Thr73 for Rab10) for all of these Rab proteins lies within the effector-binding, switch II motif (Pfeffer, 2001; Cherfils & Zeghouf, 2013). LRRK2 phosphorylation of Rab8A and Rab10 proteins blocks binding to Rab GDP-dissociation inhibitor (GDI) that is required for Rab protein membrane delivery and recycling; phosphorylation also inhibits binding of Rab8A to Rabin-8, its cognate guanine nucleotide exchange factor (GEF) (Steger *et al*, 2016, 2017). Pathogenic mutations located within the GTPase (R1441G/C) and COR (Y1699C) domains do not directly stimulate LRRK2 kinase activity *in vitro* (Jaleel *et al*, 2007; Nichols *et al*, 2010); nevertheless, they markedly enhance phosphorylation of Rab isoforms to an even greater extent than the G2019S mutation *in vivo* (Ito *et al*, 2016; Steger *et al*, 2016). These mutations promote GTP binding to the LRRK2 Roc domain (Guo *et al*, 2007; Lewis *et al*, 2007; Li *et al*, 2007; Daniels *et al*, 2011; Webber *et al*, 2011; Liao *et al*, 2014). One

1  MRC Protein Phosphorylation and Ubiquitylation Unit, School of Life Sciences, University of Dundee, Dundee, UK
2  Department of Biochemistry, Stanford University School of Medicine, Stanford, CA, USA
3  The Michael J. Fox Foundation for Parkinson's Research, New York, NY, USA
  *Corresponding author. Tel: +44 1382 385602; E-mail: f.tonelli@dundee.ac.uk
  **Corresponding author. Tel: +1 650 725 5130; E-mail: pfeffer@stanford.edu
  ***Corresponding author. Tel: +44 1382 385602; E-mail: d.r.alessi@dundee.ac.uk
  †These authors contributed equally to this work

explanation for why the R1441G/C and Y1699C LRRK2 mutants are more active in cells is that enhanced GTP binding induces a conformational change, rendering LRRK2 more susceptible to activation by an as yet unknown, upstream activator.

Rab29 is one of five genes contained within the PARK16 locus linked to Parkinson's disease (Simon-Sanchez et al, 2009; Tucci et al, 2010). It is most closely related to Rab32 and Rab38 GTPases that are needed for lysosome-related organelle biogenesis (Bultema & Di Pietro, 2013; Wang et al, 2014). In contrast to Rab29, Rab32 and Rab38 do not possess a LRRK2 phosphorylation site within their Switch II effector-binding motifs. There is significant variability within the PARK16 locus and Parkinson's association patterns across populations, and it is currently unclear how mutations within the PARK16 locus are linked to Parkinson's disease. Transcriptome analysis has suggested that the PARK16 locus enhances expression of Rab29 (Beilina et al, 2014). Other genetic studies of Parkinson's patient cohorts have found common variants within the LRRK2 and Rab29 genes that function coordinately to increase Parkinson's risk, as human genetic variants at these loci impact Parkinson's risk non-additively (MacLeod et al, 2013; Pihlstrom et al, 2015). A haplotype located near the 5′ region of RAB29 is associated with Parkinson's and epistasis between Rab29 and LRRK2 gene variants has been demonstrated (Pihlstrom et al, 2015).

Genetic investigations in Caenorhabditis elegans neurons revealed that the RAB29 (GLO-1) orthologue acts upstream of LRRK2 (LRK-1) in a signaling pathway controlling axon termination (Kuwahara et al, 2016). It was also reported that Rab29 and LRRK2 double-knockout mice exhibit an enlarged kidney phenotype that was non-additive, relative to single Rab29 or LRRK2 knockout, further implying that these genes act in a common pathway (Kuwahara et al, 2016). Others have suggested that LRRK2 and various Rab proteins including Rab29 interact, largely based on co-immunoprecipitation analysis, but binding domains have not yet been pinpointed (Dodson et al, 2012; MacLeod et al, 2013; Beilina et al, 2014; Waschbusch et al, 2014; Zhang et al, 2015). It has also been reported that Rab29 recruits LRRK2 to the Golgi apparatus (MacLeod et al, 2013; Beilina et al, 2014).

LRRK2 is constitutively phosphorylated at a cluster of Ser residues lying between the ankyrin domain and leucine-rich repeat region (Ser910, Ser935, Ser955 and Ser973) that plays a role in regulating 14-3-3 binding and cytosolic localization (Nichols et al, 2010; Doggett et al, 2011). These sites have received a lot of attention as they are controlled by LRRK2 kinase activity, and therefore become rapidly dephosphorylated in response to diverse LRRK2 inhibitors (Dzamko et al, 2010; Doggett et al, 2011). Monitoring the dephosphorylation of these residues, especially Ser935, has become the principal biomarker strategy to assess the in vivo efficacy of LRRK2 inhibitors (Hatcher et al, 2017). Despite a lot of research, it is currently unclear how LRRK2 kinase activity influences phosphorylation of these biomarker sites. Autophosphorylation would be the simplest model to account for the data obtained to date. However, autophosphorylation of the biomarker residues has not been observed in in vitro studies undertaken thus far, perhaps indicating a missing factor is required to stimulate autophosphorylation of these sites (Dzamko et al, 2010). Other kinases not known to be regulated by LRRK2 including CK1 (Chia et al, 2014) and PKA (Muda et al, 2014) have also been reported to phosphorylate these sites. In macrophages, the IkappaB kinase family phosphorylates Ser910 and Ser935 sites independently from LRRK2 kinase activity (Dzamko et al, 2012).

In this study, we demonstrate that Rab29 functions as a critical upstream regulator of LRRK2 by stimulating kinase activity, inferred by assessing autophosphorylation of Ser1292 as well as phosphorylation of LRRK2 substrates such as Rab10. We find that the pathogenic LRRK2 mutants that bind GTP with higher affinity are activated by Rab29 to a much greater extent than wild-type LRRK2, suggesting a mechanism by which such mutations stimulate LRRK2 activity in vivo. Our studies suggest that LRRK2 interacts with Rab29 via its N-terminal ankyrin domain, and, strikingly, mutations that disrupt regulation by Rab29 prevent phosphorylation of a cluster of highly studied biomarker phosphorylation sites (Ser910, Ser935, Ser955 and Ser973) (Dzamko et al, 2010; Doggett et al, 2011). Our findings suggest that Rab29 plays a major role in regulating LRRK2 Golgi localization and kinase activity as well as potentially triggering phosphorylation of biomarker sites.

## Results

### Rab29 activates LRRK2

To explore whether Rab29 influences LRRK2 kinase activity, we co-expressed Rab29 with wild-type and pathogenic mutants of LRRK2 in HEK293 cells, and assessed LRRK2 autophosphorylation of Ser1292 (Sheng et al, 2012) as well as phosphorylation of endogenous Rab10 (Thr73) and Rab29 (Thr71), employing well-characterized phospho-specific antibodies (Fig EV1). Overexpression of Rab29 significantly enhanced both LRRK2 Ser1292 and Rab10 phosphorylation (Fig 1A). Rab29 stimulated activity of the "enhanced GTP-binding mutants" (R1441C, R1441G, R1441H, Y1699C) to a much greater extent than wild-type LRRK2 (Fig 1A). Rab29 was also phosphorylated by the R1441G/C and Y1699C mutants to a much greater extent than wild-type LRRK2, consistent with the higher activation state of these pathogenic mutants (Fig 1A). Due to its higher basal activity, the G2019S mutant displayed elevated kinase activity in the absence of Rab29 overexpression, which was further enhanced upon Rab29 overexpression. The I2020T, T2031S, and G2385R pathogenic mutants behaved more like wild-type LRRK2 and were activated by Rab29 overexpression to a lesser extent than the enhanced GTP-binding mutants (Fig 1A). In general, the amount of Rab10 phosphorylation correlates with the extent of LRRK2 activation; however, some variation correlating with the level of Rab29 expression is observed. Furthermore, there are 14 Rab proteins that are phosphorylated by LRRK2 (Steger et al, 2017) and conceivably, LRRK2 mutants may have slightly different localization or preferences for diverse Rab proteins, which could also account for variation between Ser1292 phosphorylation and Rab10 phosphorylation observed (Fig 1A). We also found that stimulation of Ser1292 as well as Rab10 phosphorylation induced by overexpression of Rab29 was abolished by introducing a kinase-inactivating D2017A mutation (Fig 1A, right panels), confirming that Rab29 was enhancing phosphorylation by stimulating LRRK2 kinase activity.

To test whether phosphorylation of Rab29 at Thr71 and Ser72 by LRRK2 was required for activation of LRRK2, we mutated these sites both to Ala. We found that the Rab29[T71A,S72A] mutant still activated LRRK2[R1441G] to the same extent as wild-type Rab29, indicating that phosphorylation of Rab29 is not required for its ability to activate LRRK2 (Fig 1B). In an attempt to mimic phosphorylation of

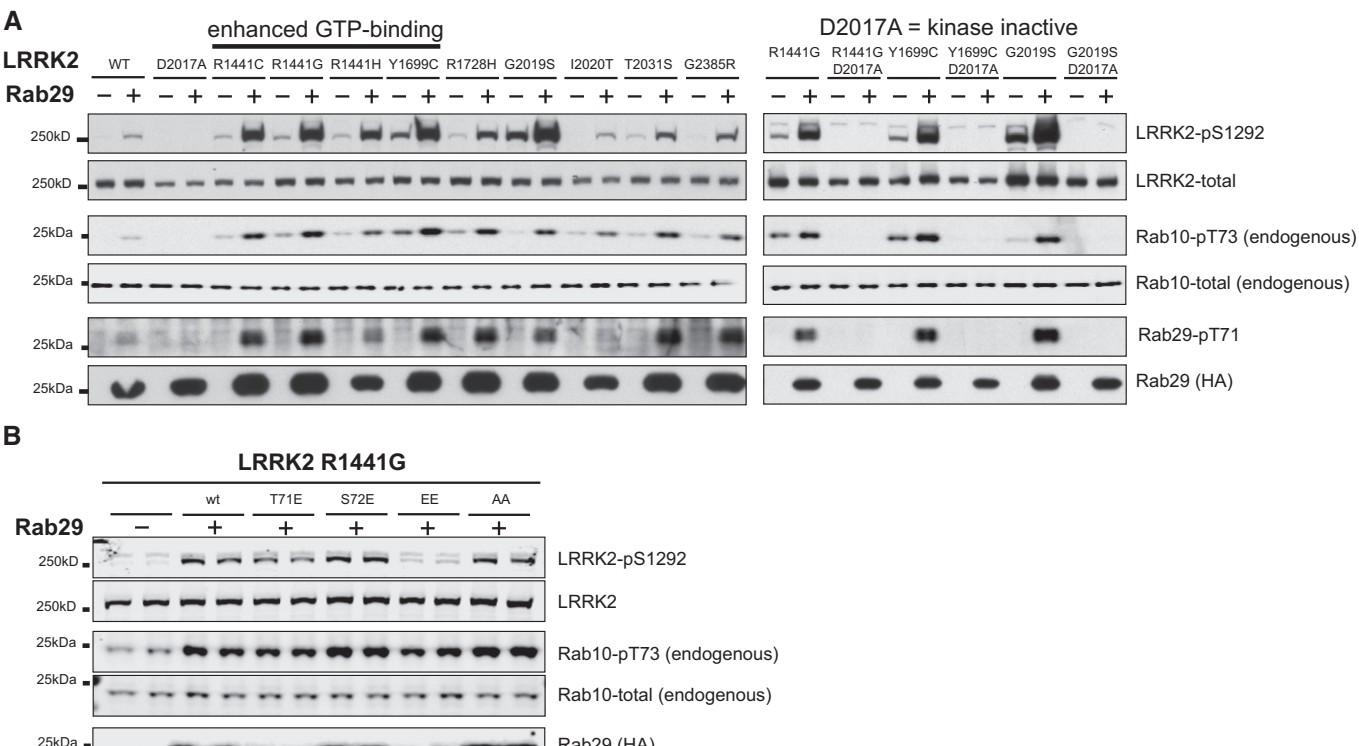

**Figure 1.  Rab29 activates LRRK2.**

A   Left: HEK293 cells were transfected with the indicated wild-type and human full-length pathogenic LRRK2 variants with either HA-empty vector (−) or HA-tagged Rab29 (+). 24 h post-transfection, cells were lysed and analyzed by immunoblotting with the indicated antibodies. WT is wild-type and D2017A corresponds to the kinase-inactive LRRK2 mutant. Similar results were obtained in two separate experiments. Right: As in left panel except that kinase-inactivating D2017A LRRK2 mutation was inserted into the indicated LRRK2 pathogenic mutant. Similar results were obtained in two independent experiments.

B   As in (A) except that LRRK2[R1441G] pathogenic variant was co-transfected with wild-type and indicated mutants of Rab29. EE indicates Rab29[T71E,S72E] mutation and AA indicates Rab29[T71A,S72A] mutation. Similar results were obtained in two separate experiments, each performed in duplicate.

Rab29 by LRRK2, we mutated the phosphorylation sites to Glu and observed that the Rab29[T71E,S72E] mutant failed to activate LRRK2 (Fig 1B). This suggests that Rab29 phosphorylation may decrease its ability to activate LRRK2.

### Rab29 selectively activates LRRK2

We next evaluated the effect that 11 Rab proteins including Rab32 and Rab38 (that are highly related to Rab29) have on Ser1292 phosphorylation of wild-type LRRK2 (Fig EV2A) and LRRK2[R1441G] (Fig EV2B). We found that for wild-type LRRK2, Rab29 markedly stimulated Ser1292 phosphorylation, but with exception of Rab12, which induced a modest ~twofold increase, none of the other Rab proteins including Rab32 and Rab38 activated LRRK2 significantly (Fig EV2A). For the LRRK2[R1441G] mutant, Rab29 also markedly increased Ser1292 phosphorylation more than any of the other Rab proteins (Fig EV2B), but Rab8A and Rab38 also stimulated Ser1292 phosphorylation two- to threefold (Fig EV2B).

### Rab29 activates LRRK2 on Golgi membranes

At steady state, LRRK2 is primarily cytosolic; approximately 10% associates with membranes upon cell fractionation. The large pool of cytoplasmic protein obscures the localization of the membrane-associated pool in fixed cells. To overcome this challenge, and to avoid the possibility of spurious precipitation of cytosolic LRRK2 protein onto cellular structures during fixation, we employed an established liquid nitrogen coverslip freeze–thaw protocol (Seaman, 2004) to deplete cytosolic proteins and reveal the localization of membrane-associated LRRK2 protein. Figure 2 shows the localization of R1441G LRRK2 in HeLa cells upon transient transfection. The protein is localized predominantly in the perinuclear region but distinct, peripheral punctae are also detected, and 60% of these also contain Rab10 protein (Fig 2A and C). In the perinuclear region, more than 40% of the LRRK2 punctae co-localized with Rab8 protein (Fig 2B and D). These findings are consistent with the fact that Rab10 and Rab8 are significant LRRK2 substrates (Steger *et al*, 2016).

Rab29 is localized to the Golgi complex (Wang *et al*, 2014) where it overlaps with p115 protein (Fig 3A). Like LRRK2 R1441G (Fig 2), LRRK2 G2019S also displays a distributed, punctate pattern when expressed on its own (Fig 3E). A small amount of perinuclear LRRK2 may co-localize with endogenous Rab29, but available antibodies were unable to label endogenous Rab29 protein. However, when LRRK2 G2019S was co-expressed in cells with Rab29, the proteins showed significant co-localization over an extended,

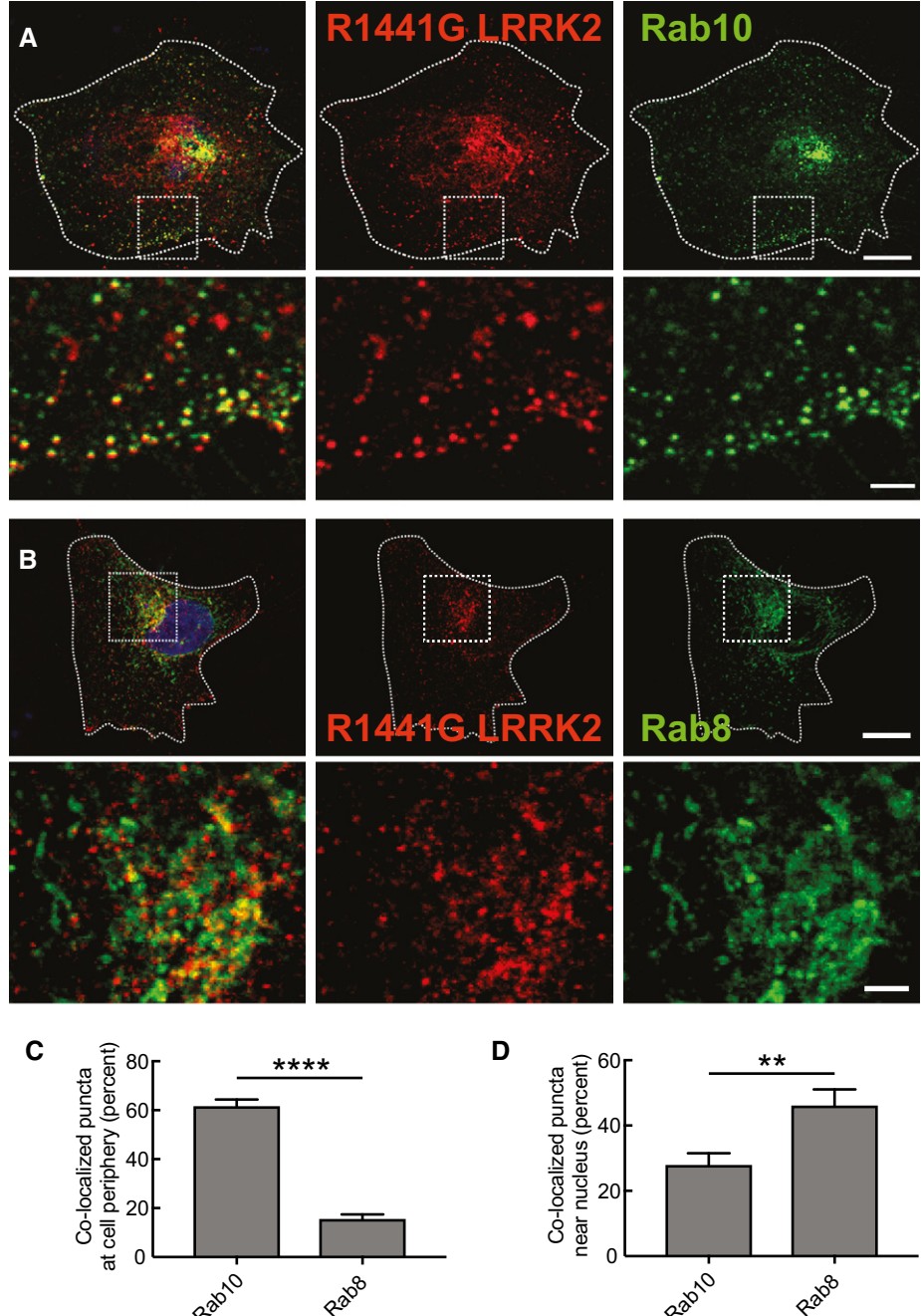

**Figure 2.  R1441G-LRRK2 co-localizes with Rab10 in the periphery and Rab8 near the nucleus.**

A, B   Shown are HeLa cells stained for transfected R1441G-LRRK2 (red) and either GFP-Rab10 (A) or GFP-Rab8 (B) in green. Scale bar, 10 μm. The second row in both panels shows enlarged portions boxed in the rows above. Scale bar, 2 μm.

C, D   Percent co-localization of R1441G-LRRK2 with the indicated Rab in the indicated cell regions was determined from a Mander's coefficient after automatic thresholding. Error bars represent mean ± SEM. ****$P < 0.0001$; **$P = 0.0076$ by Student's unpaired, two-tailed $t$-test ($n = 13$ from two experiments). For peripheral quantitation, boxes of the size indicated were generated (see A) that included the plasma membrane and excluded the nucleus. For perinuclear quantitation (see B), the boxes included half of the nucleus.

somewhat perinuclear, reticular structure (Fig 3B). This structure is likely to represent a disrupted Golgi complex, as staining became much more concentrated in the perinuclear region upon treatment of cells with the MLI-2 LRRK2 kinase inhibitor (Fig 3C and D).

Co-localization of LRRK2 G2019S (Fig 3B) and R1441G proteins (see Fig 7H below) with Rab29 supports a model in which Rab29 can activate pathogenic LRRK2 proteins on Rab29-containing membranes.

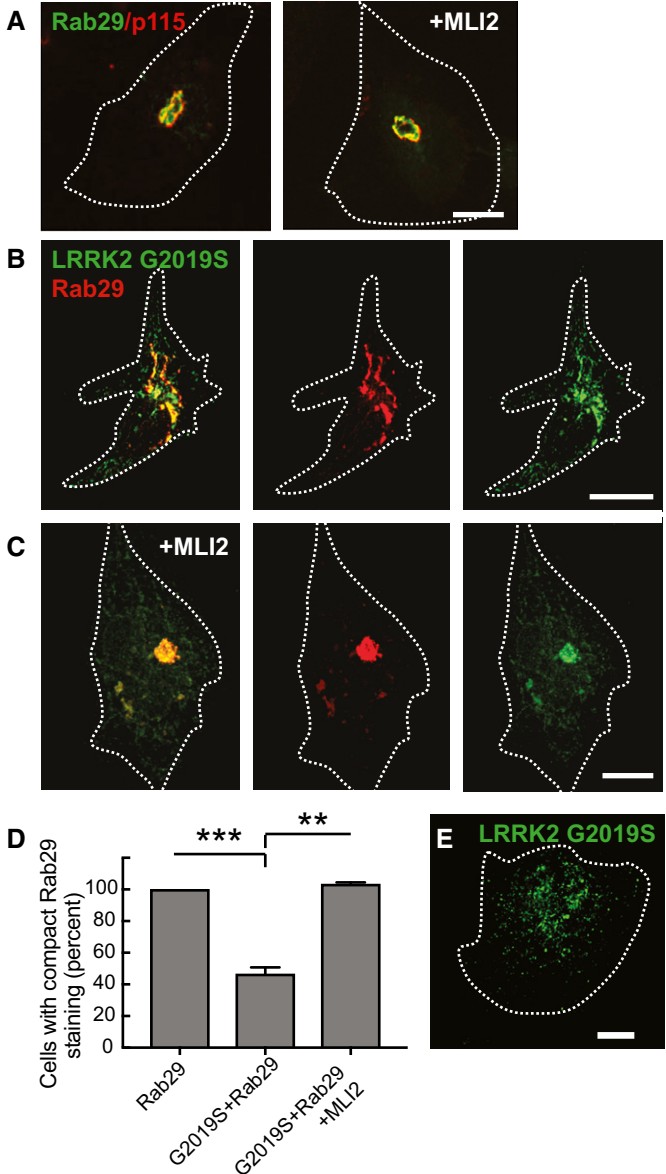

**Figure 3. Pathogenic LRRK2 mutants co-localize with Rab29 and disperse Golgi membranes.**

HeLa cells were transfected with LRRK2-G2019S or Myc-Rab29 or LRRK2-G2019S and 24 h later transfected with Myc-Rab29. After 48 h, cells were permeabilized by liquid nitrogen freeze–thaw to deplete cytosol and then fixed and stained with mouse anti-p115, mouse anti-Myc, and rabbit anti-GFP or rabbit anti-LRRK2 antibodies.

A  Myc-Rab29 (green) and p115 (red) show co-localization at the Golgi. Left and right panels were treated with or without MLI2 (200 nM, 4 h) as indicated.

B  Myc-Rab29 (red) and eGFP-LRRK2-G2019S (green) show co-localization and dispersed Rab29-labeled Golgi membranes.

C  Cells treated with MLI-2 (200 nM, 4 h) show compact, Rab29-positive Golgi, and associated LRRK2-G2019S (green).

D  Percent of cells with compact Rab29 staining; ***$P$ = 0.0002; **$P$ = 0.002 with Student's unpaired, two-tailed $t$-test. Error bars represent SEM for three experiments with > 30 cells per condition in each experiment.

E  LRRK2-G2019S alone showing punctate staining throughout the cell.

Data information: Scale bars, 10 μm.

Disruption of the Golgi by pathogenic LRRK2 proteins has been previously reported (MacLeod *et al*, 2013; Beilina *et al*, 2014), and light microscopic analysis of the *trans*-Golgi network in mouse embryonic fibroblasts (MEFs) (as monitored using anti-GCC185 antibodies) confirmed this finding (Fig 4A and C); MLI2 treatment restored compact Golgi localization for GCC185-stained compartments (compare Fig 4B and C right and left panels). Note that the extent of Golgi fragmentation correlated with the relative kinase activation of the LRRK2 mutant proteins. Together, these data show that LRRK2 co-localizes with its substrates, Rab10 and Rab8, and also co-localizes with an important key activator, Rab29 GTPase.

Rab29 activation of LRRK2 would be predicted to occur on membrane surfaces harboring active Rab29 protein. To test this, cells expressing Rab29 and R1441G LRRK2 were fractionated into membrane and cytosol fractions and analyzed for their content of total and activated LRRK2 protein, using anti-LRRK2 and anti-pS1292 antibodies. Expression of Rab29 increased the amount of total membrane-associated LRRK2 (Fig 5A and B) and also led to a greater than threefold enrichment of activated, pS1292 LRRK2 on membranes (Fig 5C). Similar data were obtained for G2019S-LRRK2 (see Fig 7 below). Note that about 10% of LRRK2-R1441G is also detected in membrane fractions obtained from Rab29 knockout 293 T cells (Fig 5B). This Rab29-independent pool may represent association of LRRK2 with another Rab GTPase such as Rab8A or Rab38, or could represent R1441G LRRK2 aggregates.

### Ankyrin domain residues permit activation of LRRK2 by Rab29

In an initial attempt to define the region of LRRK2 required for Rab29 activation, we generated a truncation mutant of LRRK2 lacking the N-terminal, 969 non-catalytic residues encompassing the armadillo and ankyrin domains. Although this mutant is expressed at lower levels than full-length LRRK2 in HEK293 cells, it was clearly not activated by Rab29 overexpression (Fig 6A), indicating that the Rab29 effector region lies within the LRRK2 N-terminal fragment, which was also suggested in a previous study (Beilina *et al*, 2014).

Rab32 and Rab38, the Rab proteins most closely related to Rab29, interact directly with the ankyrin domain of an effector called VARP (VPS9-ankyrin repeat protein, also known as ANKRD27), a regulator of endosomal trafficking (Fukuda, 2016). The crystal structure of the VARP ankyrin domain/Rab32 complex (PDB 4CYM) reveals a large interface of interacting residues, encompassing several clusters of adjacent Leu residues on VARP that make hydrophobic interactions with Rab32 (Hesketh *et al*, 2014). Mutation of these VARP ankyrin domain Leu residues decreased binding to Rab32 (Hesketh *et al*, 2014). These studies prompted us to explore whether the LRRK2 ankyrin domain might comprise the Rab29 binding site. Inspection of the LRRK2 ankyrin domain reveals that it possesses three Leu-rich motifs that are conserved in human, chicken, *Xenopus*, Zebrafish, and *Drosophila* LRRK2, which we termed Region A, B, and C (Fig 6B). We mutated representative Leu residues within each of these regions to Asp, as this corresponds to the mutations that were designed to prevent the interaction of VARP with Rab32 (Hesketh *et al*, 2014). We studied how these mutations impacted Rab29-mediated LRRK2 activation. This revealed that in particular, Region A (Cys727Asp, Leu728Asp, and Leu729Asp) mutations strikingly prevented Rab29-mediated activation of

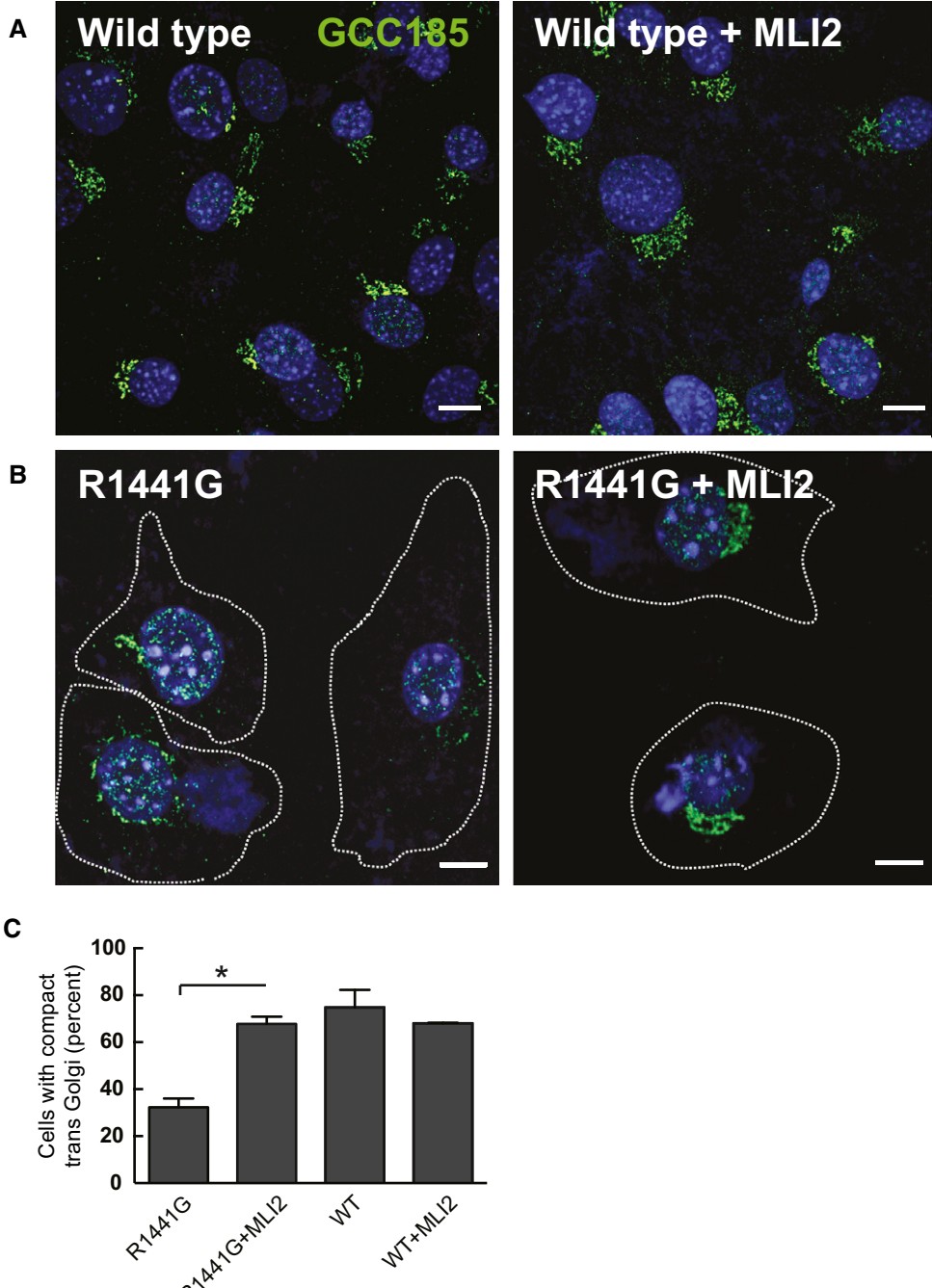

**Figure 4. LRRK2-R1441G expression disrupts Golgi morphology.**

A   Wild-type MEFs were stained with rabbit anti-GCC185 (green) and DAPI (blue); left and right panels were treated with or without MLI2 as indicated.
B   Knock-in R1441G MEF cells ± MLI-2 (200 nM, 4 h) stained with rabbit anti-GCC185 antibody (green) and DAPI (blue).
C   Quantitation of the percent of cells showing a compact *trans*-Golgi network as seen in (A). Error bars represent SEM from two experiments with > 50 cells per condition in each experiment. *$P$ = 0.0184 with Student's unpaired, two-tailed *t*-test. Differences between WT, WT+MLI2, and R1441G+MLI2 were not significant ($P$ > 0.5).

Data information: Scale bars, 10 μm.

wild-type LRRK2 (Fig 6C) and LRRK2[R1441G] mutant (Fig 6D). These ankyrin domain mutations reduced basal levels of Ser1292 phosphorylation, consistent with these enzymes being less active due to their inability to bind Rab29 (Fig 6C and D). Mutations in Region B and Region C also suppressed Rab29-stimulated Ser1292 autophosphorylation of wild-type LRRK2, further supporting a role for the ankyrin repeat domain playing a critical role in controlling LRRK2 activation (Fig 6C).

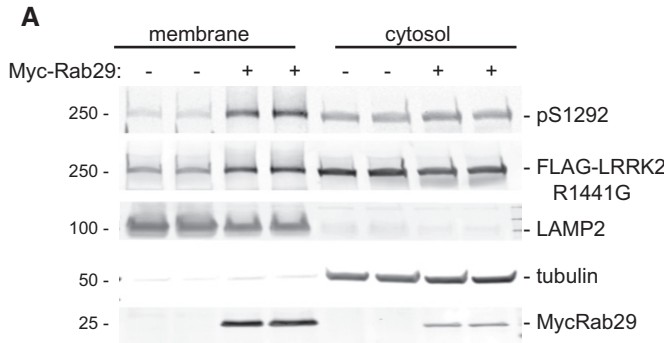

**A**

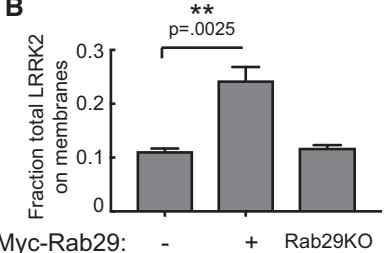

**B**

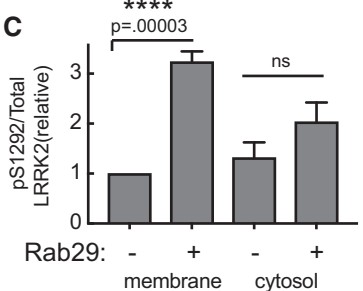

**C**

experiments demonstrated that the Region A mutations (Cys727Asp, Leu728Asp, and Leu729Asp) inhibited LRRK2 kinase activity toward Nictide and Rab8A by four- to fivefold (Fig 6E and F). At the end of *in vitro* kinase assays, we also assessed LRRK2 autophosphorylation at Ser1292 by immunoblot, which also revealed that Region A mutations also substantially inhibit autophosphorylation (Fig 6E and F).

The crystal structure of the VARP:Rab32 complex reveals that Rab32 binding to the VARP ankyrin domain is controlled by two conserved Met91 and Arg93 residues that lie within the Rab32 effector-binding switch II motif (Hesketh *et al*, 2014). Mutation of these residues to Ser abolished VARP binding (Hesketh *et al*, 2014). Interestingly, Rab29, Rab32, and Rab38 are the only Rab proteins that possess Met and Arg residues at the equivalent positions within their effector-binding loops (Fig EV3A), perhaps suggesting that this family of Rab proteins binds ankyrin domains via a common mechanism. Although we found mutation of equivalent residues in Rab29 (Met73 and Arg75) prevented activation of LRRK2 (Fig EV3B), the Rab29[M73S, R75S] mutant was localized in the cytosol, indicating that these mutations likely disrupt Rab29 nucleotide binding and C-terminal prenylation and should not be employed in future studies (Fig EV3C).

**Ankyrin domain residues influence LRRK2 membrane association and Rab29 co-localization**

If the ankyrin domain is important for Rab GTPase interaction, it would be predicted to be important for LRRK2 membrane association. To test this, cells expressing LRRK2 R1441G or LRRK2 R1441G protein also carrying Region A mutations were fractionated and analyzed for their content of membrane-associated LRRK2 protein by immunoblot. As shown in Fig 7A, approximately 10% of total R1441G LRRK2 was detected on membranes; the individual L728D or L729D LRRK2 proteins showed only a slight decrease in overall membrane association, and the L728D/L729D mutant protein led to a 60% decrease in membrane association determined by this method (Fig 7B). In addition, the ankyrin domain mutants failed to respond to Rab29 co-expression in terms of their activation on membranes, as monitored using anti-LRRK2 pS1292 antibodies (Fig 7C, E and F) or cytosol (Fig 7D); Rab29 overexpression enhanced the membrane association of total (Fig 7E) and pS1292-LRRK2 (Fig 7F) on membranes. Further support for the importance of ankyrin domain sequences in Rab29 interaction comes from light microscopy experiments, in which R1441G LRRK2 ankyrin domain mutant proteins showed significantly less co-localization with Rab29 upon expression in HeLa cells (Fig 7G and H). R1441G LRRK2 staining was much more punctate than that seen for G2019S (compare with Fig 3E); 40% of R1441G LRRK2 punctae co-localized with Rab29, and most Rab29 remained associated with a disrupted Golgi complex (Fig 7G and H). Importantly, R1441G L728D/L729D failed to disrupt the Golgi (Fig 7I), consistent with the requirement for Rab29 interaction to mediate this process. These experiments support a model in which ankyrin domain residues are important for Rab29 interaction, LRRK2 kinase activation, and Golgi complex disruption.

Ankyrin domain mutant proteins showed decreased overall kinase activity, and the protein was somewhat less stable in cells, as determined by monitoring its turnover after cycloheximide addition (Fig 7J). Loss of a binding partner interaction often leads to decreased protein stability.

**Figure 5. Rab29 increases membrane association of LRRK2-R1441G.**

HEK293T Rab29$^{-/-}$ (KO) and WT cells were transfected with LRRK2-R1441G and 24 h later transfected with Myc-Rab29. After 48 h, cells were harvested and fractionated into cytosol and membrane fractions.

A   Immunoblot of membrane protein (75 μg) and the 50% of equivalent volume of cytosolic proteins, ±Rab29 as indicated. Numbers at left indicate mobility of marker proteins in kDa; proteins were detected with rabbit anti-pS1292, rabbit anti-LRRK2 UDD3, mouse anti-LRRK2, mouse anti-LAMP2, mouse anti-tubulin, and mouse anti-Myc antibodies.
B   Quantitation of the fraction of total LRRK2-R1441G on membranes ± Rab29 (transfected and endogenous).
C   Amount of active (pS1292) LRRK2 in membrane and cytosol fractions ± Rab29 expression normalized to the amount of total LRRK2 in the fraction.

Data information: Error bars represent SEM from two experiments. **$P$ = 0.0025; ****$P$ = 0.00013; ns = not significant ($P$ = 0.1968) by Student's unpaired, two-tailed *t*-test. Differences between Rab29 KO and (minus) Myc-Rab29 are not significant.
Source data are available online for this figure.

To study further how Rab29 binding controls LRRK2, we immunoprecipitated wild-type LRRK2 and Region A mutations from HEK293 cells and assayed kinase activity employing the Nictide peptide substrate (Dzamko *et al*, 2010) (Fig 6E) or recombinant Rab8A (Steger *et al*, 2016) (Fig 6F). Assays were undertaken in the presence or absence of the MLi-2 LRRK2 inhibitor to ensure activity measured was mediated by LRRK2 rather than a contaminating kinase. These

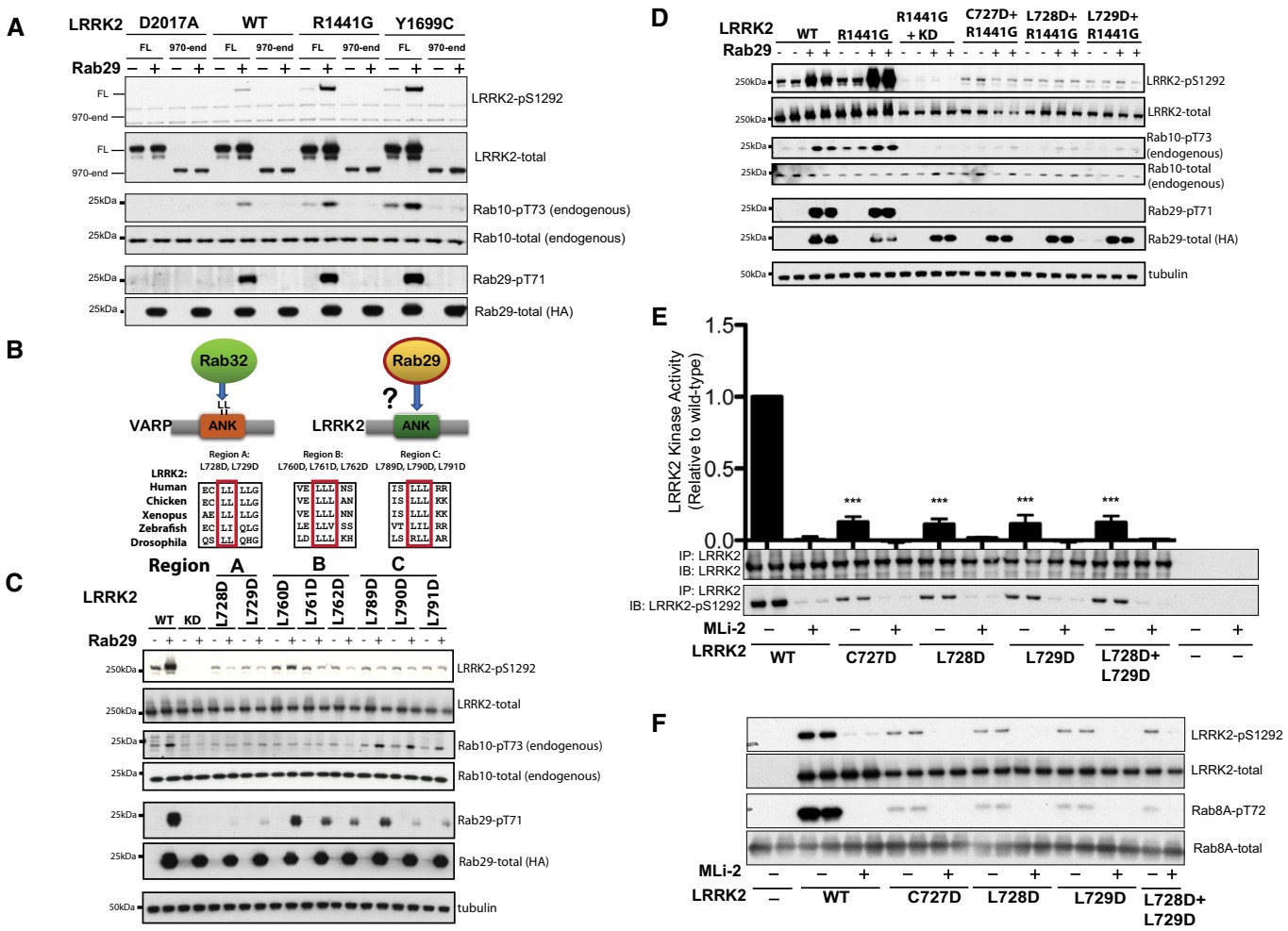

**Figure 6. Ankyrin domain residues permit activation of LRRK2 by Rab29.**

A    HEK293 cells were transfected with the indicated wild-type human full length (FL) or a fragment lacking the N-terminal 969 residues (LRRK2[residues 970-end]) pathogenic LRRK2 variants with either HA-empty vector (−) or HA-tagged Rab29 (+). 24 h post-transfection, cells were lysed and analyzed by immunoblotting with the indicated antibodies. WT is wild-type and D2017A corresponds to the kinase-inactive LRRK2 mutant. Similar results were obtained in two independent experiments, each performed in duplicate.

B    Upper panel: Schematic representation of how Rab29 might interact with the ankyrin domain (ANK) of LRRK2 by analogy with how Rab32 binds VARP. Lower panel: Sequence alignments of the three Leu-rich regions in the ankyrin domain of LRRK2 in the indicated species.

C, D    As in (A) except that HEK293 cells were transfected with the wild type and indicated LRRK2 ankyrin domain mutations with either HA-empty vector (−) or HA-tagged Rab29 (+). KD is the Kinase Dead LRRK2[D2017A] mutant. Similar results were obtained in two experiments.

E    As in (C) except that the indicated forms of LRRK2 were immunoprecipitated from cell extracts and then subjected to an LRRK2 kinase activity by measuring phosphorylation of the Nictide peptide substrate in the presence of 0.2 mM $^{32}$PγATP and in the absence (−) or presence (+) of 1 μM MLi-2 LRRK2 in a 30-min kinase reaction. After the kinase assay, phosphorylation of Nictide was quantified by Cherenkov counts and data presented as average ± SEM for three independent experiments each undertaken in triplicate. Cherenkov counts recorded for no LRRK2 (-) controls were subtracted from all values. There was a statistically significant difference between groups ($P < 0.0001$, one-way ANOVA, $F(9, 20) = 95.87$) ***$P < 0.001$ by one-way ANOVA with Dunnett's multiple comparison with mean difference 95% confidence intervals of groups compared to WT: WT MLI-2 0.8668–1.126; C727D 0.7428–1.002; C727D MLI-2 0.8731–1.132; L728D 0.7602–1.019; L728D MLI-2 0.8559–1.115; L729D 0.7570–1.016, L729D MLI-2 0.8708–1.130, L728D+L729D 0.7466–1.006; L728D+L729D MLI-2 0.8655–1.124. Assay mixtures were subjected to immunoblot analysis with the indicated antibodies.

F    As in (E) except phosphorylation of Rab8A by LRRK2 was assessed using a phospho-specific antibody. Similar results were obtained in two experiments each undertaken in duplicate.

Source data are available online for this figure.

## Evidence that Rab29 may regulate phosphorylation of LRRK2 biomarker sites

LRRK2 possesses a cluster of well-studied, constitutively phosphorylated residues (Ser910, Ser935, Ser955 and Ser973) that are controlled by LRRK2 kinase activity, as these residues become dephosphorylated following administration of LRRK2 inhibitors (Dzamko *et al*, 2010; Nichols *et al*, 2010; Doggett *et al*, 2011). We studied whether ankyrin domain mutations that prevent Rab29 from activating LRRK2 influence phosphorylation of these sites. Strikingly, all Region A

**Figure 7.   Membrane association of ankyrin domain mutant LRRK2 proteins.**

A      Immunoblots of R1441G LRRK2 in membrane (top two rows) or cytosol (bottom two rows) fractions after 48-h expression in HEK293T cells. TfR (transferrin receptor) and tubulin loading controls are included.

B      Relative membrane association of the constructs analyzed in (A). Error bars represent SEM from duplicate samples of a representative experiment. **$P$ = 0.001 using Student's unpaired *t*-test. Differences between WT and single mutations in ankyrin domain were not significant.

C, D   Membrane and cytosol fractionation of extracts from cells expressing G2019S LRRK2 or its derived ankyrin domain mutant forms, ±Rab29 as in Fig 5.

E, F   Quantitation of the relative membrane association of each construct, normalized for total LRRK2 expression. Error bars indicate SEM from duplicate determinations. ns = not significant, $P > 0.05$; (E) **$P$ = 0.003; (F) **$P$ = 0.0086 by Student's unpaired two-tailed *t*-test.

G      Quantitation of the fraction of the indicated LRRK2 proteins that co-localize with Rab29. Co-localization was measured using Mander's coefficient after automatic thresholding in FIJI. Error bars represent SEM from two experiments (15 cells); ***$P$ = 0.0002 by Student's unpaired *t*-test. Scale bar, 2 μm.

H      Light microscopy of HeLa cells transfected as in Fig 3 with LRRK2-R11441G or LRRK2-R1441G+L728D/L729D (red) and Rab29 (green). LRRK2-R1441G (red) co-localization on distinct Rab29 (green)-positive puncta representing dispersed Golgi membranes requires ankyrin domain sequences. Scale bar, 2 μm.

I      Quantitation of cells with compact Rab29 morphology; error bars represent SEM of three experiments with > 20 cells per experiment. **$P$ = 0.0011 for R1441G alone compared with Rab29 alone; **$P$ = 0.0049 for R1441G-LRRK2-L728/L729D. Differences between R1441G-LRRK2-L728/729D + Rab29 and Rab29 alone were not significant ($P > 0.5$).

J      Quantitation of the relative stability of LRRK2 R1441G protein (blue) in comparison with its ankyrin domain mutated form (red) in HEK293T cells transfected with LRRK2 R1441G or R1441G-LRRK2-L728/729D. After 24 h, cells were treated with 50 μg/ml cycloheximide for 0, 3, or 6 h. Error bars represent SEM from three combined experiments carried out in duplicate.

Source data are available online for this figure.

mutations (Cys727Asp, Leu728Asp and Leu729Asp) that disrupt Rab29-mediated activation of LRRK2 tested blocked phosphorylation of LRRK2 at Ser910, Ser935, Ser955, and Ser973 (Fig 8A).

### Rab29 knockout decreases endogenous LRRK2 activity and biomarker site phosphorylation

We generated two independent Rab29 knockout A549 cell lines employing a CRISPR/CAS9 approach (Fig 8B). Rab29 knockout markedly inhibited LRRK2-mediated phosphorylation of endogenous Rab10 by around twofold, measured with two different phospho-Rab10 monoclonal antibodies (Fig 8B). Consistent with Rab29 controlling biomarker phosphorylation sites, knockout of Rab29 moderately reduced LRRK2 phosphorylation at Ser935 and Ser973 (Fig 8B). We were unable to detect phosphorylation of endogenous LRRK2 at Ser1292 in these cells with available antibodies possibly due to low stoichiometry of phosphorylation of wild-type LRRK2 in these cells.

### GTP binding to LRRK2 is required for Rab29-mediated activation of LRRK2

To further study the role that LRRK2 GTP binding might play in controlling Rab29-mediated activation and biomarker phosphorylation, we employed the well-characterized LRRK2 T1348N mutation that blocks GTP binding (Ito *et al*, 2007; Taymans *et al*, 2011). Introduction of the T1348N mutation into wild-type, R1441G, Y1699C, and G2019S LRRK2 completely inhibited Ser1292 autophosphorylation, as well as Rab10 phosphorylation that was induced following overexpression of Rab29 (Fig 9A). Consistent with a previous report (Doggett *et al*, 2011), we also confirm that the T1348N mutation ablates phosphorylation of LRRK2 at the biomarker sites (Fig 9A). We also find that in endogenous homozygous LRRK2[T1348N] knock-in MEFs, although the mutation decreases LRRK2 expression, Rab10 phosphorylation and biomarker site phosphorylation are clearly abolished (Fig 9B).

Hilfiker and colleagues have recently analyzed the localization of a large set of LRRK2 mutant proteins and found that in about 20% of cells expressing certain mutant forms (but not wild type or G2019S), LRRK2 appears to associate with microtubules; microtubule association is enhanced upon LRRK2 inhibitor addition, a condition that is documented to enhance LRRK2 turnover (Blanca Ramirez *et al*, 2017). Using low LRRK2 expression levels and gentle release of cytosol with liquid nitrogen treatment, we have never seen microtubule association for LRRK2 G2019S or R1441G proteins. However, certain mutants appear to form what appear to be non-specific aggregates, including T1348N (Fig 9C). We believe these structures represent concentration-dependent "aggresomes" that are microtubule-associated protein aggregates—a finding that would be consistent with the previous microtubule association seen by others. Consistent with the inability of Rab29 to activate LRRK2 T1348N, co-expression with Rab29 did not increase the amount of membrane-associated LRRK2 T1348N, as determined by cell fractionation (Fig 9D). Note that the centrifugation protocol employed may pellet some LRRK2 T1348N protein aggregates that may also be present in the "membrane" fraction; nevertheless, the amount present in the fraction (16% of total LRRK2 protein) was unchanged upon Rab29 co-expression. Even though this mutant forms aggresomes, the 84% of the molecules that remain in the cytosol do not appear to be recruited to the Golgi by Rab29 GTPase (Fig 9D).

## Discussion

We have shown here that in addition to comprising an LRRK2 substrate, Rab29 operates as a master upstream regulator of LRRK2, controlling Golgi localization, kinase activity, and potentially N-terminal biomarker phosphorylation (Fig 10). We also demonstrate that pathogenic mutations such as R1441G/C or Y1699C that enhance GTP binding to the ROC domain are recruited to the Golgi apparatus and activated more efficiently than wild-type LRRK2. These observations are consistent with previous studies that have suggested that LRRK2 and Rab29 operate as part of a common signaling pathway (Dodson *et al*, 2012; MacLeod *et al*, 2013; Pihlstrom *et al*, 2015; Zhang *et al*, 2015; Kuwahara *et al*, 2016). Our work also reveals an intriguing interplay between Rab proteins and the LRRK2 kinase that also possesses a Rab-like GTPase domain. Our data are consistent with a model in which GTP binding to

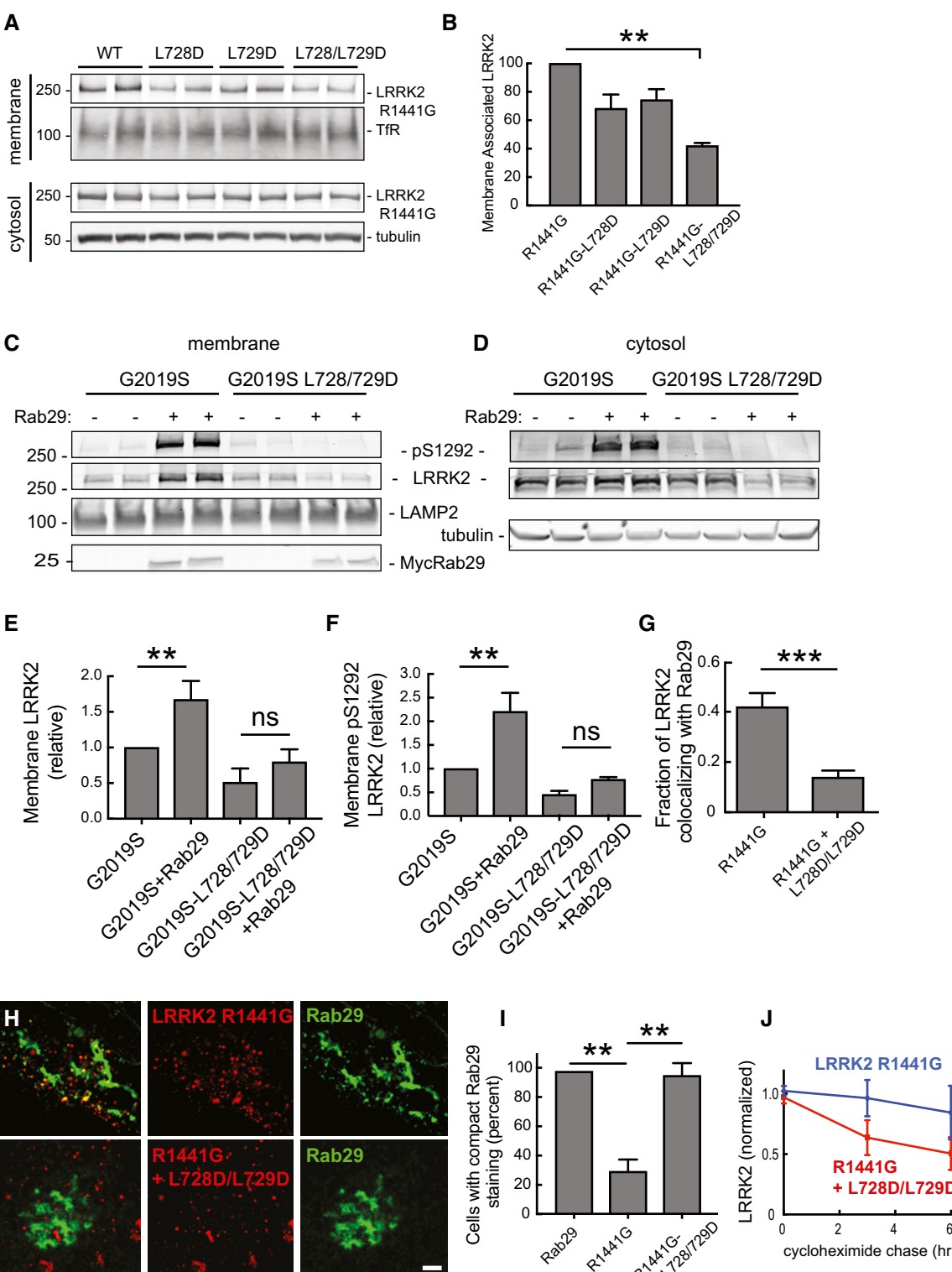

**Figure 7.**

LRRK2's ROC domain promotes Rab29 activation of LRRK2. Once activated, LRRK2 kinase phosphorylates and influences the function of a series of other downstream Rab proteins (Steger *et al*, 2016, 2017).

It was previously unclear how the LRRK2[R1441G/C] and LRRK2[Y1699C] variants activated LRRK2, as these mutants possess similar kinase activity as wild-type LRRK2 in *in vitro* experiments (Jaleel *et al*, 2007; Nichols *et al*, 2010; Steger *et al*, 2016). However, in cells,

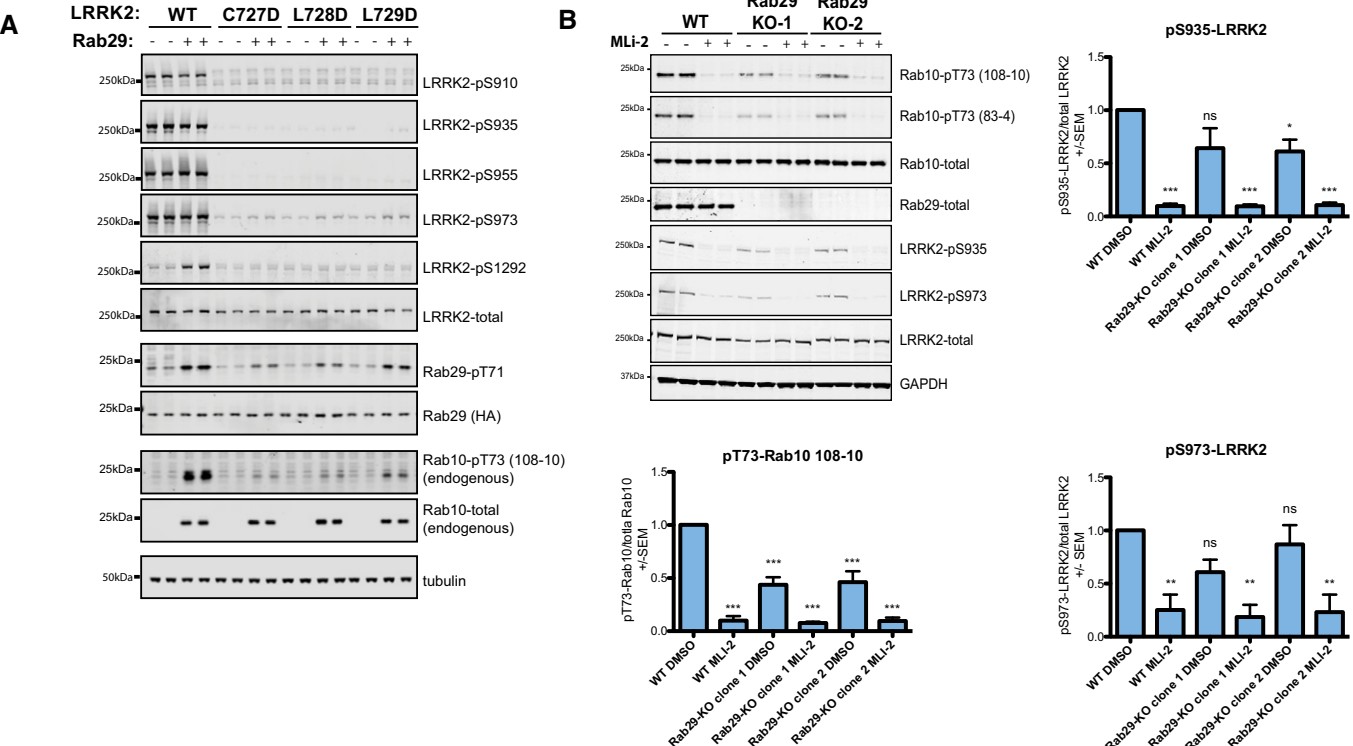

**Figure 8.  Endogenous Rab29 activates LRRK2 and is required for biomarker site phosphorylation.**

A    HEK293 cells were transfected with the indicated wild-type and ankyrin domain mutant forms of LRRK2 with either HA-empty vector (−) or HA-tagged Rab29 (+). 24 h post-transfection, cells were lysed and analyzed by immunoblotting with the indicated antibodies. WT is wild type. Similar results were obtained in two experiments.

B    Wild-type A549 cells and two independent clones of CRISPR/CAS9 Rab29 knockout (KO-1 and KO-2) were lysed and analyzed by immunoblotting with the indicated antibodies. Two separate phospho-Rab10 rabbit monoclonal antibodies were employed (108–10 and 83–4). Blots were signals quantified by LiCor and presented as average ± SEM. Similar results were obtained in three experiments. There was a statistically significant difference between groups for pT73-Rab10/total Rab10 signal ($P < 0.0001$, one-way ANOVA, $F_{(5, 12)} = 41.56$). ***$P < 0.001$ by one-way ANOVA with Dunnett's multiple comparison with mean difference 95% confidence intervals of groups compared to WT DMSO control: WT MLI-2 0.6725–1.129; Rab29-KO CL1 DMSO 0.3348–0.7916; Rab29-KO CL1 MLI-2 0.6953–1.152; Rab29-KO CL2 DMSO 0.3103–0.7671; Rab29-KO CL2 MLI-2 0.6754–1.132. There was a significant difference between groups for pS935-LRRK2/total LRRK2 signal ($P < 0.0001$, one-way ANOVA, $F_{(5, 12)} = 18.00$). $^{ns}P > 0.05$; *$P < 0.05$, ***$P < 0.001$ by one-way ANOVA with Dunnett's multiple comparison test with mean difference 95% confidence intervals of groups compared to WT DMSO control: WT MLI-2 0.5319–1.271; Rab29-KO CL1 DMSO −0.01298 to 0.7261; Rab29-KO CL1 MLI-2 0.5353–1.274; Rab29-KO CL2 DMSO 0.01820–0.7573; Rab29-KO CL2 MLI-2 0.5235–1.263. There was a significant difference between groups for pS973-LRRK2/total LRRK2 signal as well ($P = 0.003$, one-way ANOVA, $F_{(5, 12)} = 6.903$). $^{ns}P > 0.05$, **$P < 0.01$ by one-way ANOVA with Dunnett's multiple comparison test with mean difference 95% confidence intervals of groups compared to WT DMSO control: WT MLI-2 0.1955–1.300; Rab29-KO CL1 DMSO −0.1599 to 0.9449; Rab29-KO CL1 MLI-2 0.2619–1.367; Rab29-KO CL2 DMSO −0.4202 to 0.6846; Rab29-KO CL2 MLI-2 0.2159–1.321.

the LRRK2[R1441G/C] and LRRK2[Y1699C] mutations are clearly more active and phosphorylate Rab proteins to a greater extent that the LRRK2[G2019S] variant (Ito *et al*, 2016; Steger *et al*, 2016). Consistent with this, the average age of onset of Parkinson's is reportedly earlier with patients with R1441G/C mutations compared to those carrying the G2019S mutation (Gonzalez-Fernandez *et al*, 2007; Healy *et al*, 2008). Our data suggest that the LRRK2[R1441G/C] and LRRK2[Y1699C] mutations are activated *in vivo*, due to their increased ability to bind GTP, thereby promoting Rab29-mediated recruitment and activation of LRRK2 on the Golgi apparatus. Consistent with this model, introduction of the T1348N mutation that blocks GTP binding prevents Rab29-mediated recruitment of LRRK2 to the Golgi, and concomitant LRRK2 activation (Fig 9). In future work, it will be important to better define the mechanism by which the R1441G/C and Y1699C mutations promote GTP binding and how this influences Rab29 binding and LRRK2 kinase activation.

A 3D model of the structure of dimeric, full-length LRRK2 has been generated based on homology models, chemical cross-linking, negative-stain EM, and small-angle X-ray scattering (Guaitoli *et al*, 2016). Interestingly, this model indicates that the ankyrin domain lies in close proximity to the kinase domain (Guaitoli *et al*, 2016). Such an arrangement could explain the potent activation of the kinase upon Rab29 binding to the active site-adjacent ankyrin domain. Higher resolution structural analysis and further mechanistic studies are required to more precisely define how Rab29 binding to the ankyrin domain is coupled to LRRK2 kinase activity. This might also help design improved ankyrin mutants of LRRK2 that are unable to bind to Rab29 that might not destabilize the protein to better study the role that this biological interaction plays. Thus far, we have not been able to reconstitute activation of LRRK2 by Rab29 in an *in vitro* system. These experiments have been hampered by challenges in expressing fully active and mono-dispersed,

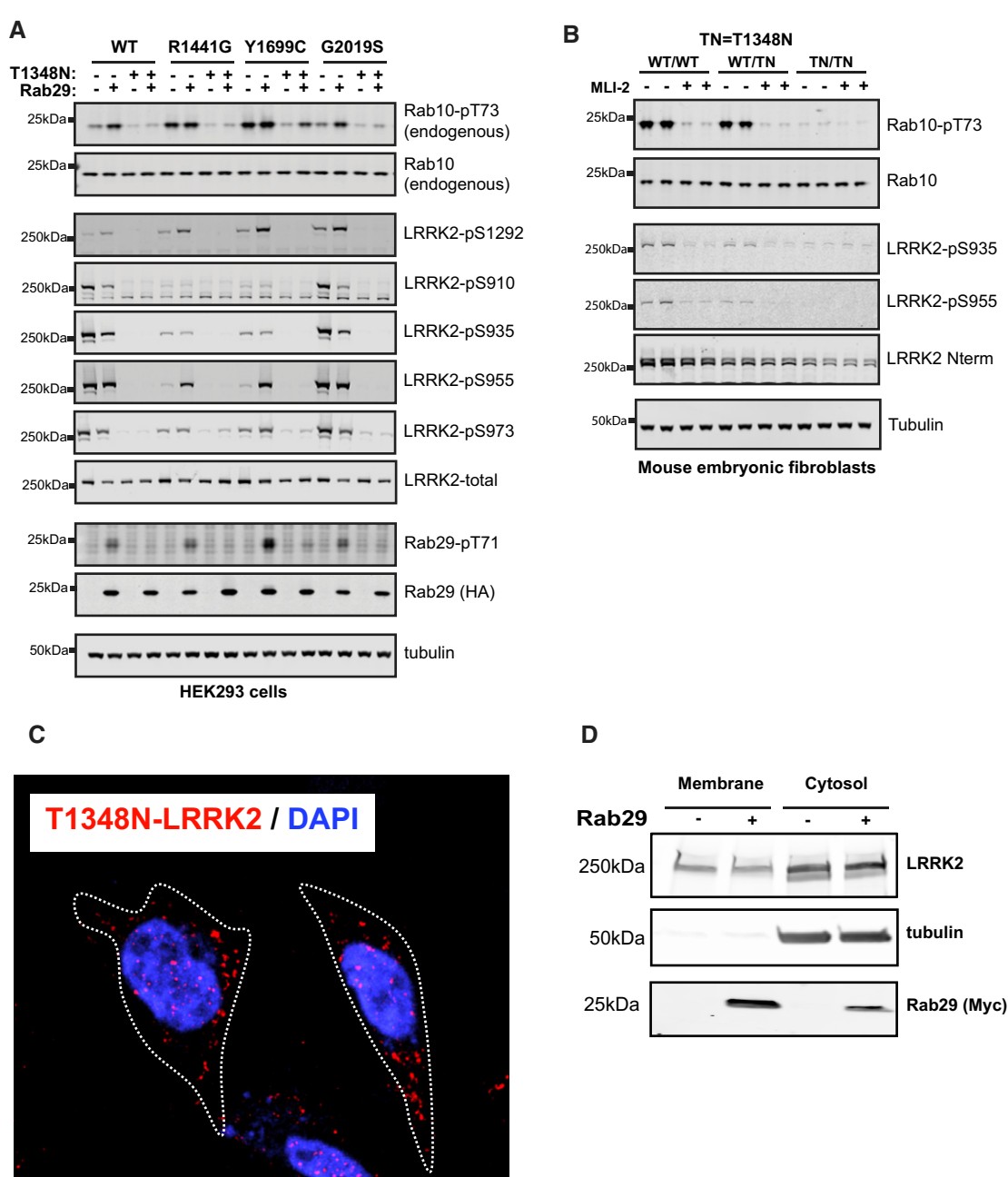

**Figure 9.  Knock-in mutation of the T1348N mutation that ablates GTP binding to LRRK2 abolishes Rab10 and biomarker phosphorylation.**

A  HEK293 cells were transfected with the indicated wild-type and mutant forms of LRRK2 with either HA-empty vector (−) or HA-tagged Rab29 (+). 24 h post-transfection, cells were lysed and analyzed by immunoblotting with the indicated antibodies WT is wild type. Similar results were obtained in two experiments.

B  Wild-type LRRK2, heterozygous LRRK2[T1348N/+], and homozygous LRRK2[T1348N/T1348N] knock-in MEFs derived from littermate embryos were lysed immunoblotted with the indicated antibodies. Similar results were obtained in two experiments. WT is Wild type and TN corresponds to T1348N.

C  HeLa cells were transfected with FLAG-T1348N-LRRK2. After 48 h, cells were permeabilized by liquid nitrogen freeze–thaw to deplete cytosol, then fixed, and stained with rabbit anti-LRRK2 antibody. Nuclear DAPI stain (blue); LRRK2 (red). Scale bar, 10 μm. Dotted line represents cell outlines.

D  HEK293T cells were transfected with FLAG-T1348N-LRRK2 and 24 h later with Myc-Rab29. After 24 h of Rab29 expression, cytosol and membrane fractions were prepared. Immunoblot of membrane protein (50 μg) and cytosolic protein (40% of equivalent volume) ± Rab29 as indicated. Numbers at left indicate mobility of marker proteins in kDa; proteins were detected with rabbit anti-LRRK2, mouse anti-tubulin, and mouse anti-Myc antibodies.

Source data are available online for this figure.

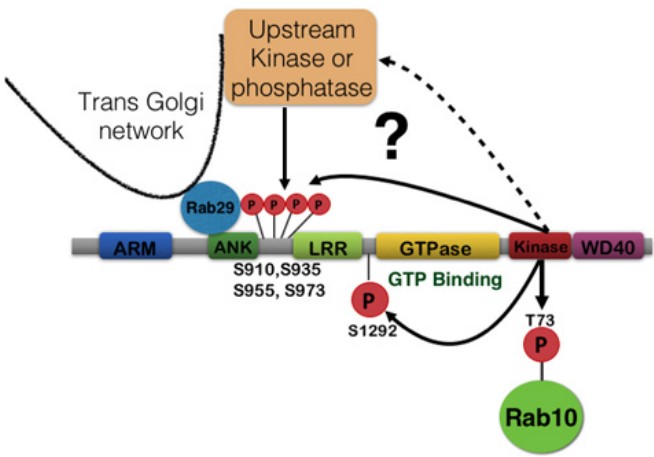

**Figure 10.  Model of how Rab29 activates and recruits LRRK2 to the *trans*-Golgi network greatly stimulating its kinase activity.**

Our data suggest that Rab29 binds to the LRRK2 ankyrin domain and that GTP binding to the ROC domain of LRRK2 promotes Rab29-mediated activation. This explains why pathogenic LRRK2 R1441G/C and Y1699C mutants that promote GTP binding are more readily recruited to the Golgi and activated by Rab29 than wild-type LRRK2. Recruitment of LRRK2 to Rab29 at the Golgi also promotes phosphorylation of a cluster of highly studied biomarker phosphorylation sites (Ser910, Ser935, Ser955, and Ser973). More work is needed to define whether these biomarker residues are phosphorylated by autophosphorylation or by a Golgi-resident upstream kinase. Finally, our data suggest that LRRK2-mediated phosphorylation of Rab29 might act as a negative feedback loop and prevent Rab29 from activating LRRK2.

recombinant Rab29. It is also possible that membrane association of Rab29 and/or other factors located on the Golgi are required for Rab29-mediated LRRK2 activation.

The present study focused on Rab29 due to the previous genetic links between Rab29, LRRK2, and Parkinson's disease. However, it is possible that other Rab proteins regulate LRRK2 localization and activity in a similar manner by binding to the ankyrin domain. Indeed, when a panel of 11 Rab proteins was tested, we observed a moderate activation of wild-type LRRK2 by Rab12 (Fig EV2A) and LRRK2[R1441G] by Rab8A and Rab38 (Fig EV2B). Recruitment of LRRK2 to membranes by different Rab proteins could comprise a general mechanism to activate LRRK2 at different locations within the cell. The Rab29-related proteins, Rab32 and Rab38, are obvious candidates for potential interactors, as reported in a recent study (Waschbusch *et al*, 2014) and indeed Rab38 can modestly activate LRRK2[R1441G] (Fig EV2A). Consistent with other Rab proteins potentially controlling LRRK2 activity, we find that in A549 cells, knockout of Rab29 significantly reduces but does not abolish LRRK2-mediated phosphorylation of Rab10 (Fig 8B). It is possible that remaining LRRK2 activity observed under these conditions is controlled by other Rab proteins binding to LRRK2.

Our data are consistent with previous work (MacLeod *et al*, 2013; Beilina *et al*, 2014), showing that recruitment of LRRK2 to the Golgi by Rab29 significantly effects Golgi apparatus integrity and induces its fragmentation in a manner that can be ameliorated by treatment with LRRK2 inhibitors (Fig 3). We also find that LRRK2 ankyrin domain mutants that are unable to interact with Rab29 do not fully disperse the Golgi apparatus, emphasizing the

necessity of Rab29 recruitment and activation for Golgi disruption (Fig 7). It will be important to determine what substrate(s) LRRK2 phosphorylates on the Golgi to trigger its disruption of the structure and whether these are additional Rab proteins. It will also be interesting to evaluate the consequences of LRRK2-mediated Golgi fragmentation in relation to Parkinson's disease. Further work is also needed to define the consequences of Rab29 phosphorylation by LRRK2. Substituting the two LRRK2 phosphorylation sites with Glu to mimic phosphorylation appears to suppress activation of LRRK2 [R1441G] (Fig 1B), potentially indicating that this could serve as a mechanism to release activated LRRK2 from the Golgi once it becomes activated. Release of Rab29-activated LRRK2 could explain its co-localization on post-Golgi structures with Rab8 in the perinuclear region and Rab10 near the periphery, two compartments that presumably lack Rab29 protein. In this regard, it is important to note that loss of Rab29 decreased Rab10 phosphorylation significantly, linking Rab29 activation with Rab10 substrate phosphorylation.

There has been much interest in how the phosphorylation of the N-terminal LRRK2 biomarker (Ser910, Ser935, Ser955 and Ser973) sites are controlled. To our knowledge, every LRRK2 kinase inhibitor tested (> 100 compounds) induces efficient dephosphorylation of these biomarker sites in around 1–2 h, suggesting that LRRK2 is somehow controlling the phosphorylation of these sites either directly through autophosphorylation or indirectly via another kinase or phosphatase (Dzamko *et al*, 2010; Doggett *et al*, 2011). Our finding that all Rab29 binding-deficient ankyrin domain LRRK2 mutants we have tested are not phosphorylated at the biomarker sites (Fig 8A), and that knockout of endogenous Rab29 in A549 cells moderately reduced phosphorylation of these sites (Fig 8B), strongly suggests that Rab29 recruitment to the Golgi is required for the phosphorylation of the biomarker sites. This is further supported by the finding that GTP binding-deficient LRRK2[T1348N] mutants that cannot be activated by Rab29, are also not phosphorylated at the biomarker sites (Fig 9A and B). This is also consistent with previous studies showing that T1348N inhibited phosphorylation of Ser935, Ser955, and Ser973 (Doggett *et al*, 2011). Overall, the data suggest that highly active LRRK2 associated with the Rab29 at the Golgi may become capable of autophosphorylation at the biomarker sites. However, our data do not exclude the possibility that another Golgi-resident, LRRK2-controlled kinase or phosphatase regulates phosphorylation of these sites. It will be vital to reconstitute activation of LRRK2 by Rab29 *in vitro* and establish whether or not this is accompanied by ability of LRRK2 to autophosphorylate at the biomarker sites.

In future work, it will also be important to study how Rab29 expression, localization, and nucleotide binding are controlled *in vivo* and to explore further, whether overexpression or activation of Rab29 is linked to Parkinson's disease. It would also be important to obtain more detailed structural information on how Rab29 binds to LRRK2. This would enable the design of improved mutants that disable binding of LRRK2 and Rab29 to better probe biological of this interaction. It would also be interesting to investigate whether Parkinson's patients with PARK16 mutations display elevated LRRK2 kinase activity and Rab10 phosphorylation. If this is the case, it would suggest that patients with PARK16 locus mutations might benefit from a future LRRK2 inhibitor therapeutic. Our data also suggest that inhibitors targeting the LRRK2 ankyrin domain

would be expected to block Rab29 binding and inhibit activity of LRRK2 in cells, thereby offering therapeutic potential for the treatment of Parkinson's disease.

# Materials and Methods

### Reagents

MLi-2 LRRK2 inhibitor (Scott *et al*, 2017) was synthesized as described in Miller *et al* (2014). All recombinant proteins, DNA constructs, and antibodies generated for the present study and more detailed information on these can be requested via our reagents website (https://mrcppureagents.dundee.ac.uk/). LRRK2 [R1441G] knock-in MEFs were kindly provided by Dr Shu-Leong Ho (Division of Neurology, Department of Medicine, University of Hong Kong, Hong Kong) and have been described previously (Ito *et al*, 2016).

### General methods

DNA constructs were amplified in *Escherichia coli* DH5α and purified using a Hi-Speed Plasmid Maxi Kit (Qiagen). DNA cloning procedures were undertaken using standard protocols. DNA sequence verification of the DNA constructs used in the present study was performed by our Sequencing Service (http://www.dnaseq.co.uk).

### Antibodies

Rabbit monoclonal antibodies for total LRRK2 (N-terminus) (UDD3) and phospho-Ser935 LRRK2 (UDD2) were generated at the University of Dundee. Mouse monoclonal antibody against total LRRK2 (C-terminus) was from NeuroMab (clone N241A/34). Rabbit monoclonal antibodies detecting phospho-Ser1292 LRRK2 [MJFR-19-7-8] (ab203181), phospho-Ser910 LRRK2 [UDD1 (15-3)] (ab133449), phospho-Ser955 [MJF-R11 (75-1)] (ab169521), and phospho-Ser973 LRRK2 [MJF-R12 (37-1)] (ab181364) were from Abcam. Anti-Rab10 total antibody was from Cell Signaling Technology (#8127) and anti-HA High Affinity (clone 3F10) from Roche. Sheep polyclonal antibody for phospho-Thr71 Rab29 (S877D) was purified at the University of Dundee and used at a final concentration of 1 μg/ml in the presence of 10 μg/ml non-phosphorylated peptide. Sheep polyclonal antibody detecting total Rab29 was purified at the University of Dundee (S984D) which can be requested via our reagents website (https://mrcppureagents.dundee.ac.uk/).

Horseradish peroxidase-conjugated anti-mouse (#31450), anti-rabbit (#31460), and anti-rat (#31470) were from Thermo Fisher Scientific. Rabbit monoclonal antibody recognizing phospho-Thr72 Rab8A/8B and phospho-Thr73 Rab10 were custom-made by Abcam in collaboration with the Michael J Fox Foundation and Abcam (Burlingame, California) (Lis *et al*, 2017). The Phospho-Rab10 antibody was raised against two phospho-T73-Rab10 peptides C-Ahx-AGQERFHT*ITTSYYR-amide (corresponds to residues 66–80 of human Rab10 in which Thr73 marked as T* is phosphorylated) and Ac-AGQERFHT*ITTSYYR-Ahx-C-amide (corresponds to residues 66–80 of human Rab10 in which Thr73 marked as T* is

phosphorylated). The phospho-Rab8 antibody was raised against two phospho-T72-Rab8A/Rab8B peptides (C-Ahx-AGQERFRT*IT-TAYYR-amide, corresponding to residues 65–79 and Ac-AGQERFR-T*ITTAYYR-Ahx-C-amide corresponding to residues 65–79 of human Rab8, *indicates the phosphorylated residue). For immunization, the peptides were coupled to KLH via the Cys residue.

### Plasmids

The following constructs were used: HA-empty vector (DU49303); HA-Rab29 wt/T71E/S72E/T71A+S72A/T71E+S72E/M73S+R75S (DU50222, DU50242, DU50243, DU52690, DU27422, DU52670, DU27495, DU27918); 6His-SUMO-Rab8a (DU47363); Flag-tagged LRRK2 wt full-length (DU6841), LRRK2 wt 970-end (DU26764), LRRK2 D2017A full-length (DU10128), LRRK2 D2017A 970-end (DU26689), LRRK2 R1441C full-length (DU13078), LRRK2 R1441G full-length (DU13077), LRRK2 R1441G 970-end (DU26770), LRRK2 R1441G+D2017A full-length (DU52702), LRRK2 R1441H full-length (DU13287), LRRK2 Y1699C full-length (DU13165), LRRK2 Y1699C 970-end (DU26763), LRRK2 Y1699C+D2017A full-length (DU52703), LRRK2 R1728H full-length (DU17138), LRRK2 G2019S full-length (DU10129), LRRK2 G2019S 970-end (DU19006), LRRK2 G2019S+D2017A full-length (DU52723), LRRK2 I2020T full-length (DU13081), LRRK2 T2031S full-length (DU17135), LRRK2 G2385R full-length (DU13083). LRRK2 C727D full-length (DU26942), LRRK2 L728D full-length (DU26916), LRRK2 L729D full-length (DU26929), LRRK2 L728D L729D full-length (DU26925), LRRK2 L760D full-length (DU27224), LRRK2 L761D full-length (DU27240), LRRK2 L762D full-length (DU27225), LRRK2 L789D full-length (DU27229), LRRK2 L790D full-length (DU27226), LRRK2 L791D full-length (DU27227), LRRK2 C727D R1441G full-length (DU27040), LRRK2 L728D R1441G full-length (DU27042), and LRRK2 L729D R1441G full-length (DU27022). Rab29 KO N-terminal antisense guide and Cas9 D10A (DU52630), Rab29 KO N-terminal sense guides (DU52626). eGFP-LRRK2-G2019S was cloned into modified pSLQ1371 with eGFP at the N-terminus. Rab29 was subcloned into pcDNA3.1 with Myc-tag and modified pSLQ1371 with eGFP at the N-terminus. All cDNA clones generated for the present study can be requested via our reagents website (https://mrcppureagents.dundee.ac.uk/).

### Purification of Rab proteins

The coding sequence for human Rab8A (accession number: NM_005370.4) was cloned into pET15b (DU47363), expressed in *E. coli* BL21 and purified as described previously (Steger *et al*, 2016).

### Cell culture, transfection, treatment, and lysis

HEK293, HeLa, A549, and mouse embryonic fibroblast cells were cultured in Dulbecco's modified Eagle's medium containing 10% fetal bovine serum, 2 mM Glutamine, and penicillin (100 U/ml)/streptomycin (100 μg/ml). Media for HEK293Trex cells before knock-in also contained 15 μg/ml Blasticidin and 50 μg/ml Zeocin. Flp-In T-REx 293 cells knock-in for eGFP-LRRK2-R1441G were maintained in 15 μg/ml Blasticidin and 100 μg/ml Hygromycin B (Thermo Scientific). LRRK2 expression was induced with 1 μg/ml

Tetracycline for 24 h. HeLa cells were transfected with Fugene 6 (Promega), and HEK293T cells were transfected with Polyethylenimine HCl MAX 4000 (Polysciences, Inc.) as described previously (Reed *et al*, 2006). Cells were lysed 24 h after transfection in an ice-cold lysis buffer containing 50 mM Tris/HCl, pH 7.5, 1% (v/v) Triton X-100, 1 mM EGTA, 1 mM sodium orthovanadate, 50 mM NaF, 10 mM 2-glycerophosphate, 5 mM sodium pyrophosphate, 0.1 μg/ml mycrocystin-LR (Enzo Life Sciences), 270 mM sucrose, and Complete EDTA-free protease inhibitor cocktail (Roche). Lysates were centrifuged at 20,800 *g* for 15 min at 4°C, and supernatants were quantified by Bradford assay (Thermo Scientific) and subjected to immunoblot analysis. Treatment of cells with MLi-2 was for 60 min at a final concentration of 100 nM, unless otherwise specified. All cell lines used in this study were tested for mycoplasma contamination and confirmed as negative for experimental analysis.

## Generation of mouse embryonic fibroblasts

LRRK2[T1348N] knock-in mice were obtained from The Jackson Laboratory and maintained on a C57BL/6J background (for further information see http://jaxmice.jax.org/strain/021829.html). Littermate-matched wild-type and homozygous LRRK2[T1348N] MEFs were isolated from mouse embryos at day E12.5 resulting from crosses between heterozygous LRRK2[T1348N/WT] mice as described previously (Wiggin *et al*, 2002). Genotyping of mice and MEFs was performed by PCR using genomic DNA isolated from ear biopsies and KOD Hot Start DNA Polymerase. Primer 1 (5′-ACAAT CATGAGCTTCATTCGGTTGTAGGGT-3′) and Primer 2 (5′-ACATAT GTGTATATAACACAACCAAGGCTGC-3′) were used to detect the wild-type and knock-in alleles. DNA sequencing was used to confirm the knock-in mutation and performed by DNA Sequencing & Services (MRC–PPU; http://www.dnaseq.co.uk) using Applied Biosystems Big-Dye version 3.1 chemistry on an Applied Biosystems model 3730 automated capillary DNA sequencer. Wild-type, heterozygous, and homozygous T1348N knock-in MEFs isolated from the same littermate were selected for subsequent experiments. Cells cultured in parallel at passage 6 were used for the immunoblotting experiments presented in this paper.

## LRRK2 immunoprecipitation kinase assays

FLAG-tagged LRRK2 wild-type and mutant variants of LRRK2 were transiently overexpressed in HEK293 cells using Polyethylenimine transfection (Reed *et al*, 2006), and 24 h post-transfection, cells were lysed in lysis buffer as described above and LRRK2 immunoprecipitated using anti-FLAG M2-agarose for 1 h (10 μl resin per 1 mg of cell extract). A control was also included where HEK293 cells were transfected with FLAG-empty vector. Immunoprecipitates were then washed three times with lysis buffer supplemented with 300 mM NaCl, and twice with 50 mM Tris/HCl (pH 7.5). Kinase assays were set up in a total volume of 50 μl with immunoprecipitated LRRK2 in 50 mM Tris/HCl (pH 7.5), 10 mM MgCl$_2$, and 1 mM ATP in the presence of 5 μg recombinant Rab8A. Assays were carried out at 30°C for 45 min with shaking. Reactions were terminated by adding LDS (lithium dodecyl sulfate) loading buffer to the beads. The mixture was then incubated at 100°C for 10 min, and the eluent was collected by centrifugation

through a 0.22-μm-pore-size Spinex column and added with 2-Mercaptoethanol to 1% (v/v). Samples were incubated for 5 min at 70°C before being subjected to SDS–PAGE and Western blotting. For the peptide substrate phosphorylation assay, kinase reactions were set up in a total volume of 50 μl with immunoprecipitated LRRK2 in 50 mM Tris/HCl, pH 7.5, 0.1 mM EGTA, 10 mM MgCl2, and 0.2 mM [γ-32P]ATP (~300–500 c.p.m./pmol) in the presence of 40 μM Nictide (RLGWWRFYTLRRARQGNTKQR). Reactions were undertaken for 30 min at 30°C and terminated by applying 45 μl of the reaction mixture on to P81 phosphocellulose paper and immersing in 50 mM phosphoric acid. After extensive washing, the radioactivity in the reaction products was quantified by Cherenkov counting. LRRK2 was then eluted from the beads by addition of LDS before being subjected to SDS–PAGE and Western blotting.

## Generation of CRISPR/Cas9 knockout of Rab29 in A549 and HEK293Trex cells

To generate Rab29 knockout cells, a modified Cas9 nickase system was used. Guides were chosen following careful transcript analysis using both NCBI and Ensembl and that the guides themselves were identified using the Sanger center's CRISPR finder (http://www.sanger.ac.uk/htgt/wge/find_crisprs). Optimal sgRNA pairs were identified with a low combined off-targeting score ((Rab29 KO-sgRNA1: GCACACTACCCAATGGAGAGC (DU52626); sgRNA2: GCTAGGTCC TGTTTCCACCTC (DU52630). Complementary oligos with BbsI-compatible overhangs were designed for each and the dsDNA guide inserts ligated into BbsI-digested target vectors; the antisense guides (sgRNA2) were cloned onto the spCas9 D10A-expressing pX335 vector (Addgene plasmid no. 42335) and the sense guides (sgRNA1) into the puromycin-selectable pBABED P U6 plasmid (Dundee-modified version of the original Cell Biolabs pBABE plasmid). A549 and HEK293Trex cells at ~80% confluency were co-transfected in a six-well plate with DU52630 and DU52626 plasmids (for the Rab29 knockout) using for A549 cells Lipofectamine LTX according to the manufacturer's instructions, with the final amount of 9 μl Lipofectamine LTX and 2.5 μg of DNA per well in a 6 well plate, and Polyethylenimine HCl MAX 4000 (Polysciences, Inc.) for HEK293Trex cells with 6 μg of Polyethylenimine, and 2.5 μg of DNA per well in a six-well plate followed by 24-h incubation in DMEM supplemented with 10% FBS, 2 mM L-glutamine, 100 units/ml penicillin, and 100 mg/ml streptomycin. Medium was then replaced with fresh medium supplemented with 2 μg/ml of puromycin. After 24 h of puromycin selection, medium was replaced again with fresh medium without puromycin and the cells were left to recover for 48 h before performing single-cell sorting.

Cell sorting was performed using Influx cell sorter (Becton Dickinson). Single cells were placed in individual wells of a 96-well plate containing DMEM supplemented with 10% FBS, 2 mM L-glutamine, 100 units/ml penicillin, and 100 mg/ml streptomycin and 100 mg/ml Normocin (InvivoGen). For HEK293Trex cells, the wells were coated by gelatin and the media contained Blasticidin and Zeocin as described above. After reaching ~80% confluency, individual clones were transferred into six-well plates. After reaching ~80% confluency, the clones were screened for presence of Rab29 by immunoblotting. A549 Rab29-KO were further confirmed by

sequencing. For this purpose, genomic DNA was isolated using GenElute™ Mammalian Genomic DNA Miniprep Kit (Sigma-Aldrich). PCR was performed using PfuUltra High-Fidelity DNA Polymerase (Agilent Technologies) using 5′-CAGGAGCGCTTCACCTCTATG and 5′-GTCTCACTCACCCTCAACATCC primers to amplify the region targeted for knockout. The PCR products were then cloned into pSC-A-amp/kan vector using StrataClone PCR Cloning Kit (Agilent Technologies). For each cloning reaction, 20 positive bacterial colonies were selected and amplified for plasmid DNA isolation using QIAprep® Spin Miniprep Kit (Qiagen). The inserts in each individual clone were then sequenced using M13 primers (DNA sequencing facility of Division of Signal Transduction Therapy at the University of Dundee) to confirm that there were no wild-type alleles of Rab29 gene present in the genome of selected clones.

## Immunoblot determination of membrane-associated LRRK2

Cells were chilled on ice, washed with ice-cold PBS, and swelled in hypotonic buffer (10 mM HEPES pH 7.4). After 15 min, 5× buffer was added to achieve a final concentration of resuspension buffer (50 mM HEPES pH 7.4, 150 mM NaCl, 5 mM MgCl$_2$, 0.5 mM DTT, 100 nM GDP, 1× protease inhibitor cocktail (Sigma) and 2 mM sodium orthovanadate, 5 mM sodium fluoride, 5 mM sodium pyrophosphate, 10 mM beta-glycerophosphate, 0.1 μg/ml Microcystin-LR), and the suspension was passed 20 times through 25-G syringe. Nuclei were pelleted by centrifugation at 1,000 *g* for 5 min at 4°C. The post-nuclear supernatant was spun 100,000 *g* for 20 min in a table top ultracentrifuge in TLA100.2 rotor; the resulting supernatant was the cytosol fraction. Membrane pellets were solubilized in 1% Triton X-100-containing 1× resuspension buffer. Protein concentrations were estimated by Bradford assay (Bio-Rad, Richmond, CA). Samples containing 50 μg of membrane protein, or the equivalent volume of cytosolic protein, were heated at 37°C for 10 min after addition of 5× SDS–PAGE sample buffer. For experiments utilizing FLAG-LRRK2-G2019S, FLAG-LRRK2-R1441G, and derivative mutants, expression was for 48 h and Myc-Rab29 expression was for 24 h. Samples were loaded in duplicate onto TGX mini-PROTEAN 4–20% precast gradient gels (Bio-Rad) or NuPAGE 3–8% precast gradient Tris-acetate gels (Invitrogen). Gels were transferred onto nitrocellulose membrane using Trans-blot turbo system, blocked in 5% milk in TBS-T. Antibodies used were rabbit anti-LRRK2 UDD3 1:1,000 (MRC PPU University of Dundee), mouse anti-Myc (Cell signaling, 1:1,000), chicken anti-GFP (Aves, 1:1,000), mouse anti-GFP (Neuromab, 1:1,000), Mouse anti-tubulin DM1A (Sigma, 1:1,000), rabbit anti-phosphoserine 1292 (Abcam, 1:500), mouse anti-LAMP2 (DSHB 1:1,000), and mouse anti-Transferrin receptor (BD Bioscience, 1:1,000). Primary antibody incubations were overnight in blocking buffer. Secondary donkey anti-Rabbit 800 and donkey anti-mouse 680 antibodies (Licor, 1:10,000) were incubated for 1 h, imaged using an Odyssey Infrared scanner (Licor), and quantified using ImageJ software.

## Light microscopy

HeLa cells were plated on collagen coated coverslips, transfected with indicated plasmids using Fugene 6 (Promega). After 48 h for LRRK2 and 24 h for Rab expression, cells were cytosol depleted by liquid nitrogen freeze-thaw (Seaman, 2004). Briefly, cells were chilled on ice, washed twice with cold PBS, and incubated in glutamate buffer (25 mM KCl, 25 mM HEPES pH 7.4, 25 mM magnesium acetate, 5 mM EGTA, 150 mM potassium glutamate). Excess buffer was removed by blotting with a tissue; the coverslip was then dipped in liquid nitrogen for 5 s and allowed to thaw for few seconds. Coverslips were then gently washed with glutamate buffer and rehydrated for 5 min in cold PBS. Cells were then fixed (cold 3% PFA for 20 min on ice), permeabilized with 0.1% Trixon X-100, and blocked with 2% BSA in PBS. Antibodies were diluted as follows: anti-LRRK2 UDD3 antibody (1:1,000, MRC PPU University of Dundee), mouse anti-GFP (Neuromab, 1:1,000), and mouse anti-Myc (9E10 Hybridoma culture supernatant—undiluted). Secondary antibodies (Thermo Scientific) were goat anti-rabbit Alexa 568 (1:2,000), goat anti-mouse Alexa 488 (1:1,000), goat anti-mouse Alexa 555 (1:2,000), and goat anti-rabbit Alexa 488 (1:2,000). Images were acquired using a laser scanning confocal microscope (Leica SP8) fitted with a 63× 1.4NA objective and acousto-optical beam splitter with hybrid detector, or a spinning disk confocal microscope (Yokogawa) with an electron multiplying charge-coupled device (EMCCD) camera (Andor, UK) and a 100× 1.4NA oil immersion objective. Images were analyzed using Fiji (https://fiji.sc/). Co-localization was quantified using JACoP, a Fiji plugin, and Mander's coefficients were calculated using an automatic threshold. MEF-R1441G or WT cells were fixed (3% PFA, RT, 15 min), permeabilized using Triton X-100 (0.1%), blocked with 2% BSA, and stained with rabbit anti-GCC185 (1:1,000), (Cheung *et al*, 2015) and goat anti-rabbit Alexa 488 (1:1,000) antibody. Nuclei were stained using 0.1 μg/ml DAPI (Sigma).

## Statistics

Graphs were made using Graphpad Prism 5 software. Error bars indicate SEM. Student's *t*-test was used to test significance. Two-tailed *P*-values < 0.05 were considered statistically significant. Figures EV2, 6E and 8B were analyzed using one-way ANOVA with Dunnett's multiple comparison test to test significance.

## Data availability

All primary data are available to anyone requesting this. We confirm that this study does not contain protein, DNA, RNA sequence, macromolecular structure, crystallographic, functional genomic, or proteomic data that are subject to the "EMBO Data Deposition policy".

**Expanded View** for this article is available online.

## Acknowledgements

We thank Dr. Martin Steger (Department of Proteomics and Signal Transduction, Max Planck Institute of Biochemistry, Martinsried, Germany) as well as Kalpana Merchant (TransThera Consulting), Marco Baptista, and Shalini Padmanabhan (Michael J Fox Foundation for Parkinson's research) for helpful discussions. We also thank Thineskrishna Anbarasan for performing CRISPR/Cas9 knockouts of Rab29 in HEK293Trex cells, Philip Wing-Lok Ho and Shu-Leong Ho (Division of Neurology, Department of Medicine, University of Hong Kong, Hong Kong) for provision of the LRRK2[R1441G] knock-in MEFs and the excellent technical support of the MRC-Protein Phosphorylation and Ubiquitylation Unit (PPU) DNA Sequencing Service (coordinated by Nicholas Helps), the MRC PPU tissue culture team (coordinated by Laura Fin), MRC

PPU Reagents and Services antibody purification teams (coordinated by Hilary McLauchlan and James Hastie). This work was supported by the Michael J. Fox Foundation for Parkinson's research [grant number 6986 (to S.R.P. and D.R.A.)]; the Medical Research Council [grant number MC_UU_12016/2 (to D.R.A.)]; the pharmaceutical companies supporting the Division of Signal Transduction Therapy Unit (Boehringer-Ingelheim, GlaxoSmithKline, and Merck KGaA, to D.R.A.); and the U.S. National Institutes of Health DK37332 (to S.R.P.).

## Author contributions

EP designed and executed experiments (Figs 6B–D and 8A, and 9A and B, and EV2, and EV3A and B); HSD designed and executed experiments (Figs 2 and 3 and 4 and 5 and 7A–I, and 9C and D, and EV3C); FT designed and executed experiments (Figs 1, and EV1A, and 6A, E and F) and was the first to discover that Rab29 stimulated LRRK2 Ser1292 phosphorylation; ARS executed experiment Fig 8B; RG executed experiment Fig 7J; PL generated Rab29 KO A549 cells and helped generate the phospho Rab8 and Rab10 antibodies and executed experiment Fig EV1B and C; MW undertook most of the cloning studies; TNM designed, organized, and oversaw the effort required for development the rabbit phospho Rab8 and Rab10 antibodies; SRP and DRA supervised the project and wrote the manuscript. All authors were involved in discussing and interpreting the data.

## Conflict of interest

The authors declare that they have no conflict of interest.

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
