## [Review Process File · The EMBO Journal]

Rab29 activation of the Parkinson's disease-associated LRRK2 kinase

Elena Purlyte, Herschel S. Dhekne, Adil R Sarhan, Rachel Gomez, Pawel Lis, Melanie Wightman, Terina N. Martinez, Francesca Tonelli, Suzanne R. Pfeffer & Dario R. Alessi

Review timeline:

Submission date:	27 August 2017
Editorial Decision:	20 September 2017
Revision received:	22 October 2017
Editorial Decision:	7 November 2017
Revision received:	13 November 2017
Accepted:	16 November 2017

Editor: Hartmut Vodermaier

Transaction Report:

1st Editorial Decision

20 September 2017

Thank you for submitting your manuscript on Rab29-LRRK2 interplay to The EMBO Journal. It has now been seen by three expert referees, whose reports are copied below for your information. Since all referees consider the study potentially interesting and important, we would be happy to consider it further for publication, pending satisfactory revision in light of the reviewers' comments. I should note that while most of the points raised may probably be addressed in a rather straightforward manner, referee 2 indicates that deeper follow-up work on the cellular significance of the described biochemical findings would be desirable; while I agree that any such data you may have would indeed broaden the ramifications of this work, I realize that carefully addressing the specific points of refs 1 and 2 (especially those regarding Figure 8), and looking a somewhat more at cellular phenotypes as suggested by referee 2, would already go a long way in addressing this concern.

I would like to remind you that it is our policy to allow only a single round of major revision, making it important to carefully respond to all points raised at this stage. Should you have any additional questions/comments regarding the referee reports or the revision requirements, please therefore do not hesitate to get in touch with me ahead of resubmission. If needed, we might also extend the revision period, during which publication of any competing work elsewhere would have no negative impact on our final assessment of your own study.

Please refer to the sections below for additional information on preparing, formatting and uploading a revised manuscript.

Thank you again for the opportunity to consider this work for The EMBO Journal. I look forward to your revision.

 REFEREE REPORTS

Referee #1:

This manuscript explores the interaction of LRRK2 with Rab29. Prior studies show that LRRK2 binds Rab29, but have not explored the role of Rab29 in LRRK2 function, or the interactions between Rab29 and mutant LRRK2 forms. The manuscript adds a great deal of extra knowledge about these interactions. They demonstrate that the R1441G/C and Y1699C LRRK2 mutants exhibit enhanced phosphorylation and recruitment to the Golgi in response to Rab29. This is an interesting story that provides a significant advance in our knowledge of the regulation of LRRK2. The manuscript, though, has some major and minor weaknesses that need shoring up. These are described below.

Major criticisms:

1. Fig. 1A: The results should note that Rab29 enhanced activity of GTP binding mutants more than other mutants FOR THE pS1292 site. However, enhancement of Rab10 pT73 phosphorylation was similar for all of the mutations. On the other hand enhancement of the Rab29 pT71 phosphorylation showed more mutation related variability, with mutations like I2020 giving little phosphorylation.
2. Fig. 8: The KO studies in figure 8B are very important. Since the response of phospho-S1292 to Rab29 over-expression is presented prominently in Fig. 1A, the authors should determine whether Rab29 KO inhibits S1292 phosphorylation. Also, since the authors present control of LRRK2 localization by Rab29 as a major axis of regulation, they should determine whether loss of Rab29 is required for control of LRRK2 localization or membrane binding of WT or R1441 mutant LRRK2.
3. The methods for cryo-permeabilization are not described, nor is there any demonstration on the degree of membrane vs cytoplasmic concentration.
4. Fig. 2: The protocol for selecting perinuclear and peripheral puncta are not described, opening up the possibility that the selection was prone to selection bias. Please provide an objective protocol and that can provide confidence that the quantification is objective.
5. Fig. 5: The immunoblot for Fig. 5A, pS1292 looks quite weak. Can this be improved? Fig. 5B: I doubt that 40% of total LRRK2 is at the membrane. This does NOT reflect the immunoblot. The quantification of pS1292 cytosol showing a 50% increase with Rab29 is also questionable.

Minor criticisms:

6. All the bar graphs need stars in the figures, showing which changes are statistically significant.
7. MLI2 is not defined on first use.
8. Figure 4: The ML12 cells still don't look like the WT or WT/ML12 cells. Either they should explain the difference or they should pick a more representative picture.

Referee #2:

The manuscript by Purlyte et al follows up on exciting recent data from the same group reporting that LRRK2 phosphorylates a series of Rab proteins. Here they report the highly novel finding that Rab29 serves as a universal and upstream activator of LRRK2 kinase activity - regardless of missense mutations in LRRK2. Rab29-mediated activation of LRRK2 is associated with its membrane localization and appears to involve N-terminal residues/structure within the ankyrin repeat domain. These conclusions are further bolstered by the observation that KO of endogenous Rab29 reduces basal activity of endogenous LRRK2.

Most of the results shown are very clear, robust and well described. There is a clear effect not previously reported and of high significance to the LRRK2 field. The manuscript may identify a possible explanation for the mechanism underlying the non-G2019S mutations causal for PD. However, the manuscript is also very narrowly focused on biochemical activation of LRRK2 without substantial new insight into the biological consequence of these biochemical interactions (what is the function of LRRK2? How is activity at the Golgi influence the Golgi?) in over-expressed or endogenous systems. While function may be a tall order, cellular phenotypes with respect to WT LRRK2 and how LRRK2 mutations changes these, would be expected for this particular journal. While the scope of the current manuscript is limited to a narrow biochemical observation, this group can certainly address these issues and improve the impact of the manuscript. There are a range of some major and mostly minor concerns

1. To be suitable for publication, attention to statistical analysis is required. There is a blurb in methods describing the statistical methods used, but there were no statistics applied anywhere in the paper. Experiments were performed only twice throughout (often in duplicate) but it would appear that the histograms are frequently of the single experiment (representative) and not pooled across all samples. Here, some of the effects are borderline and it is not clear how to interpret results, especially without analysis of all replicates (e.g. effects of Rab29 KO on LRRK2 phosphorylation) and greater repetition as well as confirmation in multiple cell types. Many of the effects are deserving of greater replication and such formal analysis, given their borderline changes and the importance of the experiments themselves to the model. Too few cells, puncta or replicates are applied and in the absence of stats to fully appreciate all the work that was conducted (e.g. Figs 2, 3, 4, 5, 7, 8)

2. This manuscript follows up on the prior work from this group demonstrating LRRK2-dependent phosphorylation of Rabs, and identifies the upstream partner in stimulating this process. While the manuscript adds biochemical insight it does not add biological insight beyond this. The cellular efforts are disjointed. The authors choose the R1441G mutation to study co-localization with Rab8 and Rab10, G2019S for co-localization with Rab29, and WT and R1441G over-expression (without untransfected cells) for "compact Golgi". Changing constructs and conditions diminishes what we can learn about 1) what is WT LRRK2 doing (include untransfected, KD etc) in the cell and 2) how is this biology influenced by its mutation. These data are the closest to LRRK2 mechanism and in their current form are a great missed opportunity to advance the understanding of what happens after LRRK2 is activated.

3. The authors highlight that Rabs 32 and 38 were the inspiration for how Rab29 might activate LRRK2. However, these are never used as negative controls for LRRK2 activators - negative and positive controls are absent throughout, but these are particularly evident. Its not clear if the authors have identified one of many LRRK2 activators or a very novel relationship between these two - this could substantially improve impact. Since Rab29 is both an activator and substrate, the other Rab LRRK2 substrates must also be ruled out as co-activators, as well - its possible that over-expression of a substrate is sufficient to activate the over-expressed kinase - a simple experiment that can quickly be ruled out.

4. It is not adequately described/studied whether the "Region A" like mutations are simply detrimental to LRRK2 folding and function. They clearly abrogate Rab29-mediated activation, but they also reduce the basal activity of LRRK2 substantially. While one cannot really compare across cell types, these mutations appear stronger than Rab29 KO, suggesting that the effects are not just via inhibition of Rab29 binding. Much more care must be paid here to figure out the results. This is a bit glossed over in the manuscript, only focusing on the differences in the presence of Rab29 - the differences in the absence of Rab29 are just as profound/novel/surprising.

5. There are problems with Figure 8 where the role of endogenous Rab29 is sought to be confirmed. While pRab10 levels are reduced, many other "LRRK2 biomarkers" are not, and not in lock step with the pRab10. Since LRRK2 is likely not the only Rab10 kinase, and LRRK2 may also be one of many proteins activated by Rab29 - its not clear here that the effects of Rab29 KO are LRRK2-dependent. As in above, these discrepancies are not sufficiently pinned. p935 is not really reduced and p1292 is not studied. As stated above, without statistical analyses of normalized pRab10/total Rab10 and p935/total LRRK2 one cannot draw conclusions from the data as provided.

Minor Comments

-The exploration of phosphomimetic and alanine mutations in Rab29 are thoughtful - its interesting to note that phosphorylation might detach Rab29 from LRRK2 and limit LRRK2 activation. I am not sure what other experiments could be done to validate this model, but the data are important to the model

-The authors have apparently made efforts to reconstitute an in vitro Rab29/LRRK2 activation system and reported difficulties - while this would be an excellent addition this reviewer recognizes the potential challenges here and would suggest this is not necessary given the in situ data shown, as they are sufficient in this initial report.

All in all this is a strong paper from an outstanding team that makes important new biochemical inroads in the understanding of LRRK2 - however greater insight into the biological significance of these findings would improve impact and interest outside the narrow LRRK2 focused subset of parkinson disease researchers. If this is a central and biologically important process relevant to Golgi function, than all of the tools and systems are in place to demonstrate this to interest the broad readership of EMBO.

Referee #3:

Purlyte and coworkers present exciting new data on the role of the small G-protein Rab29 (Rab7L1) in the recruitment of LRRK2 to specific endomembranes in the trans Golgi network.

The present work establishes Rab29 (Rab7L1), which is localized in a PD-risk locus, as a LRRK2 docking site. The authors could show that LRRK2 binding to membrane-bound Rab29 is a prerequisite for Rab8a/10 phosphorylation, previously identified as LRRK2 substrates by the same group. In addition, the work suggests that LRRK2-mediated Rab29 phosphorylation is part of a negative feedback-loop. Furthermore, the authors demonstrate that the LRRK2 Ankyrin domain is crucial for effective Rab29-mediated membrane localization of LRRK2. Even single point mutations lead to a strong reduction of LRRK2 membrane localization and subsequent phosphorylation of well-established biomarker sites within LRRK2, including the auto-phosphorylation site S1292 as well as phospho-S910/S935. In addition, using cellular models, the authors demonstrate that pathogenic LRRK2 variants, which show increased GTP binding, augment the phosphorylation of Rab proteins, including Rab29.

The manuscript represents sound work and is clearly written, being certainly interesting for the field and the readership of the EMBO journal.

I just have two minor comments:

1. Given the observation that Rab29 is a weak LRRK2 in vitro substrate, the presented data do not entirely rule out that the observed Rab29 phosphorylation in cells is indirect.
2. The material and methods part still contains proofreading remarks (i.e. on pages 26 and 28). These should be removed and missing information should be provided.

We thank the reviewers for their thoughtful and constructive comments and respond to each point here:

Referee #1: This manuscript explores the interaction of LRRK2 with Rab29. Prior studies show that LRRK2 binds Rab29, but have not explored the role of Rab29 in LRRK2 function, or the interactions between Rab29 and mutant LRRK2 forms. The manuscript adds a great deal of extra knowledge about these interactions. They demonstrate that the R1441G/C and Y1699C LRRK2 mutants exhibit enhanced phosphorylation and recruitment to the Golgi in response to Rab29. This is an interesting story that provides a significant advance in our knowledge of the regulation of LRRK2. [THANK YOU!] The manuscript, though, has some major and minor weaknesses that need shoring up. These are described below.

Major criticisms:

1. Fig. 1A: The results should note that Rab29 enhanced activity of GTP binding mutants more than other mutants FOR THE pS1292 site. However, enhancement of Rab10 pT73 phosphorylation was similar for all of the mutations. On the other hand enhancement of the Rab29 pT71 phosphorylation showed more mutation related variability, with mutations like I2020 giving little phosphorylation.

In general, the amount of p-Rab10 correlates with the extent of LRRK2 activation; Most of the variation seems to correlate with the level of Rab29 expression and it is also important to keep in mind that these blots reflect a snapshot of rapidly turning over phosphosites. There are also multiple Rab proteins that are known to be phosphorylated by LRRK2. Conceivably LRRK2 mutants may have slightly different localization or preferences for diverse Rab proteins, which could also contribute towards variation between Ser1292 phosphorylation and Rab10 phosphorylation observed. We have now carefully revised the text of the Result section that discusses Fig 1A data based on the Reviewers comments.

2. Fig. 9: The KO studies in figure 9B are very important. Since the response of phospho-S1292 to Rab29 over-expression is presented prominently in Fig. 1A, the authors should determine whether Rab29 KO inhibits S1292 phosphorylation. Also, since the authors present control of LRRK2 localization by Rab29 as a major axis of regulation, they should determine whether loss of Rab29 is required for control of LRRK2 localization or membrane binding of WT or R1441 mutant LRRK2.

We agree with the Reviewer that it would be important to measure endogenous LRRK2 Ser1292 phosphorylation in wild type and Rab29 knock-out A549 cells. We have therefore attempted to undertake this experiment on several occasions by immunoprecipitating LRRK2 from 5 mg of cell extract. However, we were unable to detect endogenous Ser1292 phosphorylation on the low levels of wild type LRRK2 that are present in these cells. The available phospho-Ser1292 antibodies are well known to be not very sensitive. Furthermore, stoichiometry of phosphorylation of wild type LRRK2 at Ser1292 is very low compared to pathogenic mutants, as can also be seen from data presented in Fig 1A. We have now stated in text referring to Fig 9A that we were unable to detect phosphorylation of endogenous Ser1292 LRRK2 in these cells with available antibodies possibly due to low stoichiometry of phosphorylation of wild type LRRK2 in these cells. We have also recently initiated a new project with the Michael J Fox Foundation to raise a more sensitive rabbit monoclonal Ser1292 phospho-specific antibody that we hope will be significantly more sensitive than the available antibody.

We have measured the percent of membrane associated R1441G-LRRK2 pS1292 in 293T cells (containing low endogenous Rab29) and 293T cells with Rab29 knock out and in both cases, the much more abundant exogenous LRRK2 shows 10% membrane association. Upon co-expression of Rab29, this goes to 30% membrane association (new KO data included in revised Fig. 6). This may be due to intrinsic membrane localization/activity and/or activation by another Rab protein(s).

We have not succeeded in measuring localization of endogenous LRRK2 in cell types we have studied types owing to its low levels of expression. This has proved very difficult and other researchers in the field have also struggled with this. The exogenous G2019S and R1441G mutant proteins are not normally present on the Golgi unless we co-express Rab29 (Revised Fig. 4). We have clarified the text to explain this more clearly.

3. The methods for cryo-permeabilization are not described, nor is there any demonstration on the degree of membrane vs cytoplasmic concentration. The method of permeabilization is identical to that cited, but we have added more details in response to this referee concern. We do show membrane versus cytosol fractionation in Figs. 6B,C, and 8A-F.

4. Fig. 3: The protocol for selecting perinuclear and peripheral puncta are not described, opening up the possibility that the selection was prone to selection bias. Please provide an objective protocol and that can provide confidence that the

quantification is objective. We have provided the requested information in the methods section; the interpretation is meant to provide a general impression rather than an absolute distinction.

5. Fig. 6: The immunoblot for Fig. 6A, pS1292 looks quite weak. Can this be improved? Fig. 6B: I doubt that 40% of total LRRK2 is at the membrane. This does NOT reflect the immunoblot. The quantification of pS1292 cytosol showing a 50% increase with Rab29 is also questionable. As discussed above (point 2), the anti-pS1292 antibody is not a sensitive antibody and stoichiometry of phosphorylation especially of the wild type protein is low. Nevertheless, we re-did the experiment yet another time with more material and get the same (now darker) results--and have replaced the blots in Fig. 6 as requested.

Minor criticisms: 6. All the bar graphs need stars in the figures, showing which changes are statistically significant. CORRECTED 7. ML12 is not defined on first use. CORRECTED

8. Figure 5: The ML12 cells still don't look like the WT or WT/ML12 cells. Either they should explain the difference or they should pick a more representative picture. CORRECTED--it was mostly a contrast issue for green staining over a black background.

Referee #2: The manuscript by Purlyte et al follows up on exciting recent data from the same group reporting that LRRK2 phosphorylates a series of Rab proteins. Here they report the highly novel finding that Rab29 serves as a universal and upstream activator of LRRK2 kinase activity - regardless of missense mutations in LRRK2. Rab29-mediated activation of LRRK2 is associated with its membrane localization and appears to involve N-terminal residues/structure within the ankyrin repeat domain. These conclusions are further bolstered by the observation that KO of endogenous Rab29 reduces basal activity of endogenous LRRK2. Most of the results shown are very clear, robust and well described. There is a clear effect not previously reported and of high significance to the LRRK2 field. The manuscript may identify a possible explanation for the mechanism underlying the non-G2019S mutations causal for PD. However, the manuscript is also very narrowly focused on biochemical activation of LRRK2 without substantial new insight into the biological consequence of these biochemical interactions (what is the function of LRRK2? How is activity at the Golgi influence the Golgi?) in over-expressed or endogenous systems. While function may be a tall order, cellular phenotypes with respect to WT LRRK2 and how LRRK2 mutations changes these, would be expected for this particular journal. While the scope of the current manuscript is limited to a narrow biochemical observation, this group can certainly address these issues and improve the impact of the manuscript. There are a range of some major and mostly minor concerns

1. To be suitable for publication, attention to statistical analysis is required. There is a blurb in methods describing the statistical methods used, but there were no statistics applied anywhere in the paper. Experiments were performed only twice throughout (often in duplicate) but it would appear that the histograms are frequently of the single experiment (representative) and not pooled across all samples. Here, some of the effects are borderline and it is not clear how to interpret results, especially without analysis of all replicates (e.g. effects of Rab29 KO on LRRK2 phosphorylation) and greater repetition as well as confirmation in multiple cell types. Many of the effects are deserving of greater replication and such formal analysis, given their borderline changes and the importance of the experiments themselves to the model. Too few cells, puncta or replicates are applied and in the absence of stats to fully appreciate all the work that was conducted (e.g. Figs 3, 4, 5, 6, 8, 9)

As requested we have carefully documented the statistical analysis to bolster our conclusions throughout, and provide much more detail as requested. We have also repeated Fig 1B with duplicate samples shown in each lane.

2. This manuscript follows up on the prior work from this group demonstrating LRRK2-dependent phosphorylation of Rabs, and identifies the upstream partner in stimulating this process. While the manuscript adds biochemical insight it does not add biological insight beyond this. The cellular efforts are disjointed. The authors choose the R1441G mutation to study co-localization with Rab8 and Rab10, G2019S for co-localization with Rab29, and WT and R1441G over-expression (without untransfected cells) for "compact Golgi". Changing constructs and conditions diminishes what we can learn about 1) what is WT LRRK2 doing (include untransfected, KD etc) in the cell and 2) how is this biology influenced by its mutation. These data are the closest to LRRK2 mechanism and in their current form are a great missed opportunity to advance the understanding of what happens after LRRK2 is activated.

We apologize if the cellular efforts seemed disjointed and this was not our intention and they were added to support the biochemistry data. The LRRK2[R1441G] mutant disorders the Golgi significantly thus making it much harder to know what one is looking at for the Rab29 co-localization (as seen comparing G2019S in Fig 4B to R1441G in Fig 5B and Fig 8H). Moreover, Fig 5 data on the Golgi come from wild-type mouse embryonic

fibroblasts or those that carry an endogenous locus knock-in at R1441G – and thus is not overexpression. Rab29 is already known to be on the Golgi (shown in Fig 4A) and we have used it as a Golgi marker as well in Fig. 4 (with G2019S) versus GCC185 as the Golgi marker in Fig. 5 (with R1441G). We have clarified the text accordingly.

We do not yet know the normal role for LRRK2, and the cellular findings (Rab29 activation; phosphorylation of downstream Rabs) point the way for future experiments. R1441G localization is also presented later as part of the ANK mutation analysis and that mutation was used as it provides the most sensitive condition to monitor the consequences of ANK domain mutations. Regarding LRRK2 KO: Inhibition of LRRK2 has no effect on the Golgi (Fig. 5A and new panel in Fig. 4). Finally, we agree that the downstream consequences of LRRK2 activation are of great interest and we and others are working diligently to uncover them.

3. The authors highlight that Rabs 32 and 38 were the inspiration for how Rab29 might activate LRRK2. However, these are never used as negative controls for LRRK2 activators - negative and positive controls are absent throughout, but these are particularly evident. It's not clear if the authors have identified one of many LRRK2 activators or a very novel relationship between these two - this could substantially improve impact. Since Rab29 is both an activator and substrate, the other Rab LRRK2 substrates must also be ruled out as co-activators, as well - it's possible that over-expression of a substrate is sufficient to activate the over-expressed kinase - a simple experiment that can quickly be ruled out.

We thank the Reviewer for suggesting this interesting experiment that we have now undertaken. We have now evaluated the effect that 11 Rab proteins including Rab29, Rab32 and Rab38 have on both wild type LRRK2 as well as LRRK2[R1441G] Ser1292 phosphorylation (see new Fig 2). For wild type LRRK2, Rab29 markedly stimulated Ser1292 phosphorylation and with the exception of Rab12, which induced a modest ~2-fold increase in Ser1292 phosphorylation, no other Rab proteins including Rab32 and Rab38 had a significant effect (Fig 2A). For the LRRK2[R1441G] mutant, Rab29 increased Ser1292 phosphorylation much more than any of the other Rab proteins (Fig 2B). Rab8A and Rab38 were also observed to induce a moderate 2-3-fold activation stimulation of Ser1292 phosphorylation (Fig2B). These findings are discussed in the Results and Discussion. It is possible that other Rab proteins can bind to the ANK or other domains of LRRK2 and stimulate membrane recruitment of activity. It is also possible that other Rab proteins could influence levels or ability of Rab29 to activate LRRK2 by unknown mechanisms.

4. It is not adequately described/studied whether the "Region A" like mutations are simply detrimental to LRRK2 folding and function. They clearly abrogate Rab29-mediated activation, but they also reduce the basal activity of LRRK2 substantially. While one cannot really compare across cell types, these mutations appear stronger than Rab29 KO, suggesting that the effects are not just via inhibition of Rab29 binding. Much more care must be paid here to figure out the results. This is a bit glossed over in the manuscript, only focusing on the differences in the presence of Rab29 - the differences in the absence of Rab29 are just as profound/novel/surprising.

The reviewer is correct, and Region A mutations are likely to have additional consequences for LRRK2 protein--activity and membrane association. In new data (new Fig. 8I) we show that the protein is less stable than the parental LRRK2 R1441G protein in cells. We have carefully modified the text to clarify this point. It is not unusual for proteins to be less stable when unable to bind to an important partner (in this case Rab29). Furthermore, without an available structure of the LRRK2-Rab29 complex, we have attempted to do our best to disrupt binding based on available structural information of how the ANK domain of VARP binds to Rab32. For region A, we study 4 different point mutants (C727D, L728D, L729D and L728D+L729D) that all show similar results. In addition, we have also tested six other mutations in the Region B & C site, and find that many of these also partially inhibit Rab29 mediated activation of LRRK2 (Fig 7C). Future structural studies are needed to characterize the Rab29 binding site and this will undoubtedly enable the design of a better set of mutants to dissect the role that Rab29 binding to LRRK2 plays in biology.

5. There are problems with Figure 9 where the role of endogenous Rab29 is sought to be confirmed. While pRab10 levels are reduced, many other "LRRK2 biomarkers" are not, and not in lock step with the pRab10. Since LRRK2 is likely not the only Rab10 kinase, and LRRK2 may also be one of many proteins activated by Rab29 - it's not clear here that the effects of Rab29 KO are LRRK2-dependent. As in above, these discrepancies are not

sufficiently panned. p935 is not really reduced and p1292 is not studied. As stated above, without statistical analyses of normalized pRab10/total Rab10 and p935/total LRRK2 one cannot draw conclusions from the data as provided.

We feel that the overall conclusion of the figure supports the view that Rab29 promotes phosphorylation of the biomarker sites. Based on the Reviewers comments, we have carefully modified the text in order not to overstate our data. We discuss that biomarker site phosphorylation could be triggered through a Rab29 mediated LRRK2 autophosphorylation mechanism or via another Golgi resident kinase or phosphatase. We do our best to illustrate these models in summary Fig 11 and include a “?” in the figure to highlight that the mechanism is not solved. We also now state in the abstract that Rab29 potentially controls biomarker phosphorylation. We also employ the MLI-2 LRRK2 specific inhibitor in Fig 9B, to highlight the LRRK2 specific effects. We also feel that the findings described in Fig 10, showing that the T1348N mutation which prevents GTP binding to LRRK2 also stops Rab29 activation as well as biomarker site phosphorylation, provides further support to the conclusion that Rab29 and likely other Rabs play a role in regulating biomarker phosphorylation sites. Given that there is so much interest in the biomarker phosphorylation sites and companies are using these sites as pharmacodynamic markers for characterizing LRRK2 inhibitors, we would very much prefer to retain this data in the paper.

Minor Comments

-The exploration of phosphor-mimetic and alanine mutations in Rab29 are thoughtful - its interesting to note that phosphorylation might detach Rab29 from LRRK2 and limit LRRK2 activation. I am not sure what other experiments could be done to validate this model, but the data are important to the model

-The authors have apparently made efforts to reconstitute an in vitro Rab29/LRRK2 activation system and reported difficulties - while this would be an excellent addition this reviewer recognizes the potential challenges here and would suggest this is not necessary given the in situ data shown, as they are sufficient in this initial report. We thank the referee for acknowledging that full reconstitution will require significant additional work and is beyond the scope of the present story

All in all this is a strong paper from an outstanding team that makes important new biochemical inroads in the understanding of LRRK2 - [THANK YOU] however greater insight into the biological significance of these findings would improve impact and interest outside the narrow LRRK2 focused subset of Parkinson disease researchers. If this is a central and biologically important process relevant to Golgi function, than all of the tools and systems are in place to demonstrate this to interest the broad readership of EMBO. There is so much work here and we hope the reviewer will appreciate that the biological consequences of LRRK2 phosphorylation are beyond the scope of the present story.

Referee #3: Purlyte and coworkers present exciting new data on the role of the small G-protein Rab29 (Rab7L1) in the recruitment of LRRK2 to specific endomembranes in the trans Golgi network. The present work establishes Rab29 (Rab7L1), which is localized in a PD-risk locus, as a LRRK2 docking site. The authors could show that LRRK2 binding to membrane-bound Rab29 is a prerequisite for Rab8a/10 phosphorylation, previously identified as LRRK2 substrates by the same group. In addition, the work suggests that LRRK2-mediated Rab29 phosphorylation is part of a negative feedback-loop. Furthermore, the authors demonstrate that the LRRK2 Ankyrin domain is crucial for effective Rab29-mediated membrane localization of LRRK2. Even single point mutations lead to a strong reduction of LRRK2 membrane localization and subsequent phosphorylation of well-established biomarker sites within LRRK2, including the auto-phosphorylation site S1292 as well as phospho-S910/S935. In addition, using cellular models, the authors demonstrate that pathogenic LRRK2 variants, which show increased GTP binding, augment the phosphorylation of Rab proteins, including Rab29.

The manuscript represents sound work and is clearly written, being certainly interesting for the field and the readership of the EMBO journal. [THANK YOU!]

I just have two minor comments:

1. Given the observation that Rab29 is a weak LRRK2 in vitro substrate, the presented data do not entirely rule out that the observed Rab29 phosphorylation in cells is indirect. We agree and have noted this in the text. The inefficient phosphorylation of Rab29 in vitro (reported in Steger et al., 2016) might be due to challenges in expressing fully active and mono-disperse, recombinant Rab29 in E.coli. Our current model is that Rab29 binds to the ANK domain to activate the kinase, which phosphorylates OTHER Rab GTPases via the kinase domain.
2. The material and methods part still contains proofreading remarks (i.e. on pages 26 and 28). These should be removed and missing information should be provided. We have corrected this

Thank you for submitting your revised manuscript for our consideration. Referees 1 and 2 have now assessed it once more, and generally consider most originally raised concerns adequately addressed. There are however two remaining issues related to the revision work which I feel would need to be further clarified prior to publication. In particular, the issue regarding phospho-biomarker detection in Rab29 loss-of-function cells reiterated by referee 1 would in my view require decisive clarification. Similarly, I feel that the second point about testing one of the LRRK2 mutants to strengthen the phenotypic follow-up appears well taken and would certainly elevate the impact of the study further. I would therefore kindly ask you to address these two seemingly straightforward requests through an additional round of (minor) revision.

I am therefore returning the manuscript to you for an additional round of revision, hoping that you will be readily able to satisfactorily respond to the remaining points. Please do not hesitate to get back to me should you have any further questions.

REFEREE REPORTS

Referee #1:

This is a resubmitted manuscript demonstrating that Rab29 controls activation of LRRK2, as demonstrated by increased phosphorylation of LRRK2 biomarker pS1292 and a corresponding phosphorylation of Rab10 pT73. The manuscript is also significant for identifying a specific action of the LRRK2 R1441G/C and Y1699C mutants (that promote GTP binding), which answers a long-standing question in the field. The authors responded to many of the criticisms well, adding statistics, modifying the text and clarifying use of replicates.

Originally in the manuscript I identified a potential weakness in the logic of the manuscript because the authors used the biomarker pS1292 throughout the manuscript, EXCEPT for the knockdown experiments where they used the biomarkers pS935/973. I pointed this out. The authors addressed the point, by saying that they tried to detect pS1292 in the A549 cells but couldn't. This response is noted, and I agree that phosphorylation patterns could differ among cell lines. This response seems odd to me because the methods section explicitly states that Rab29 was knocked out in A549 cells AND HEK293 tRex cells. Because Figure 1 definitely shows that endogenous levels of pS1292 are detectable in HEK cells, I am concerned about the absence of experiments examining the effects of Rab29 KO in the HEK cells. My concern is that the effects of Rab29 on LRRK2 phosphorylation represent an "over-expression artifact". The authors need to show some biomarker that is elevated in figure 1 (perhaps by additionally examining pS935/973?) and reciprocally reduced in Figure 9 (Perhaps by examining HEK Rab29 KO cells?). If the HEK Rab29 KO cells do not actually exist, the authors should correct the wording of the methods section AND do knockdown experiments in HEK cells to accomplish this task. Absence of such reciprocal proof suggests an inability to accomplish the task, and the possibility that this is an over-expression artifact.

Reviewer #2 raised an important point that for EMBO journal the authors need to show a phenotypic outcome associated with regulation of LRRK2 by Rab29. I agree with this point. To this end, I note experiment 4C demonstrating a consolidation of the Golgi apparatus with the LRRK2 inhibitor MLI2. Golgi condensation strikes me as a reasonable phenotype to assess, however inhibition with MLI2 is not sufficient proof that the action is due to the Rab29/LRRK2 axis (it could be due to effects on proteins other than Rab29, or it could be due to off-target effects of MLI2). However, assuming that the phenomenon does reflect the Rab29/LRRK2 axis, it seems relatively straight forward to test the effects of one of the mutants from figure 9 (C/L727, 728, 729 or A935 or 973), and test for Golgi consolidation in HEK cells using a Golgi marker. This would provide a "physiological readout".

I realize that my review requires a second revision, but since the experiments all utilize rapidly and easily cultured HEK cells, these experiments do not strike me as difficult tasks. The rest of the manuscript is well done. Should the authors be able to achieve these revisions, they would have an outstanding article that would represent a strong addition to the field.

Referee #2:

The authors have addressed each point with as much diligence as is currently possible with acknowledged technical limitations - the revision should be considered state of the art. The new data added provide even deeper insight into the complicated interplay between Rabs and LRRK2 and will make an impactful and novel contribution to the field.

2nd Revision - authors' response

13 November 2017

This is a resubmitted manuscript demonstrating that Rab29 controls activation of LRRK2, as demonstrated by increased phosphorylation of LRRK2 biomarker pS1292 and a corresponding phosphorylation of Rab10 pT73. The manuscript is also significant for identifying a specific action of the LRRK2 R1441G/C and Y1699C mutants (that promote GTP binding), which answers a long-standing question in the field. The authors responded to many of the criticisms well, adding statistics, modifying the text and clarifying use of replicates.

Originally in the manuscript I identified a potential weakness in the logic of the manuscript because the authors used the biomarker pS1292 throughout the manuscript, EXCEPT for the knockdown experiments where they used the biomarkers pS935/973. I pointed this out. The authors addressed the point, by saying that they tried to detect pS1292 in the A549 cells but couldn't. This response is noted, and I agree that phosphorylation patterns could differ among cell lines. This response seems odd to me because the methods section explicitly states that Rab29 was knocked out in A549 cells AND HEK293 tRex cells.

Because Figure 1 definitely shows that endogenous levels of pS1292 are detectable in HEK cells, I am concerned about the absence of experiments examining the effects of Rab29 KO in the HEK cells. This is a **misunderstanding** as Fig. 1 shows pS1292 of EXOGENOUS proteins; note here that even exogenous *wild type* LRRK2 has a very weak pS1292 signal. The levels of overexpression of wild type and mutant LRRK2 in this experiment is much higher than that of endogenous Rab29. Because of this Rab29 knockout in HEK293 cells does not affect level activity of overexpressed LRRK2 (we have done the experiment). The HEK293 cell experiment in Fig 1 is a model system that reveals activation of LRRK2 by Rab29

Moreover, In Figure 6B, we expressed LRRK2 R1441G in HEK293 wild type or Rab29 KO cells. We detected the same level of membrane association of LRRK2 seen without exogenous Rab29 and much less than that seen with exogenous Rab29 expression. This experiment demonstrates Rab29-dependent membrane association of LRRK2 and indicates that some LRRK2 is also membrane associated in a Rab29-independent manner. Moreover, overexpression of Rab29 M73S R75S fails to activate LRRK2 (Fig. 7 expanded view panel B), showing that it relies on a functional Rab29 protein.

There is virtually no endogenous LRRK2 in HEK293 cells. This is why we work with A549 cells to study endogenous LRRK2 and In Fig 8B we demonstrate that knock-out of endogenous Rab29 inhibits Rab10 phosphorylation that is mediated by LRRK2

My concern is that the effects of Rab29 on LRRK2 phosphorylation represent an "over-expression artifact". The authors need to show some biomarker that is elevated in figure 1 (perhaps by additionally examining pS935/973?) and reciprocally reduced in Figure 9 (Perhaps by examining HEK Rab29 KO cells?). If the HEK Rab29 KO cells do not actually exist, the authors should correct the wording of the methods section AND do knockdown experiments in HEK cells to accomplish this task. Absence of such reciprocal proof suggests an inability to accomplish the task, and the possibility that this is an over-expression artifact.

Reviewer #2 raised an important point that for EMBO journal the authors need to show a phenotypic outcome associated with regulation of LRRK2 by Rab29. I agree with this point. To this end, I note experiment 4C demonstrating a consolidation of the Golgi apparatus with the LRRK2 inhibitor MLI2. Golgi condensation strikes me as a reasonable phenotype to assess, however inhibition with MLI2 is not sufficient proof that the action is due to the Rab29/LRRK2 axis (it could be due to effects on proteins other than Rab29, or it could be due to off-target effects of MLI2). However, assuming that the phenomenon does reflect the Rab29/LRRK2 axis, it seems relatively straight forward to test the effects of one of the mutants from figure 9 (C/L727, 728, 729 or A935 or 973), and test for Golgi consolidation in HEK cells using a Golgi marker. This would provide a "physiological readout".

As requested, we show that LRRK2 R1441G ANK L728/729D fails to cause Golgi fragmentation, a phenotypic readout of the importance of Rab29-LRRK2 interaction (see new Fig 7H & I).

Thank you for submitting your re-revised manuscript for our consideration. I have now looked through your responses and final version, and I am happy to inform you that we have accepted it for publication in The EMBO Journal.

Corresponding Author Name: Suzanne Pfeffer Francesca Tonelli and Dario Alessi
EMBO J
Manuscript Number: EMBOJ-2017-98099